# Evidence for microbial iron reduction in the methanic sediments of the oligotrophic SE Mediterranean continental shelf

Hanni Vigderovich[1], Lewen Liang[2], Barak Herut[3], Fengping Wang[2], Eyal Wurgaft[1,4], Maxim Rubin-Blum[3] and Orit Sivan[1]

[1]The Department of Geological and Environmental Sciences, Ben-Gurion University of the Negev, Beer-Sheva, 8410501, Israel.
[2] School of Life Sciences and Biotechnology, Shanghai JiaoTong University, Shanghai, 200240, P.R.China.
[3] Israel Oceanographic and Limnological Research, Haifa, 31080, Israel.
[4]Currently: The Department of Marine Chemistry and Biochemistry, Woods-Hole Oceanographic Institution, Woods-Hole, USA
*Correspondence to*: Orit Sivan (oritsi@bgu.ac.il)

**Abstract.** Dissimilatory iron reduction is probably one of the oldest types of metabolisms that still participates in important biogeochemical cycles, such as of the carbon and sulfur. It is one of the more energetically favorable anaerobic microbial respiration processes and is usually coupled to the oxidation of organic matter. Traditionally this process is thought to be limited to the shallow part of the sedimentary column in most aquatic systems. However, iron reduction has also been observed in the methanic zone of many marine and freshwater sediments, well below its expected zone, occasionally accompanied by decreases in methane, suggesting a link between the iron and the methane cycles. Nevertheless, the mechanistic nature of this link (competition, redox or other) has yet to be established, and has not been studied in oligotrophic shallow marine sediments. In this study we present combined geochemical and molecular evidences for microbial iron reduction in the methanic zone of the oligotrophic Southern Eastern (SE) Mediterranean continental shelf. Geochemical pore-water profiles indicate iron reduction in two zones, the uppermost part of the sediment, and the deeper zone, in the layer of high methane concentration. Results from a slurry incubation experiment indicate that the deep methanic iron reduction is microbially mediated. The sedimentary profiles of microbial abundance and qPCR of the *mcrA* gene, together with Spearman correlation between the microbial data and Fe(II) concentrations in the pore-water, suggest types of potential microorganisms that may be involved in the iron reduction via several potential pathways: $H_2$ or organic matter oxidation, an active sulfur cycle or iron driven anaerobic oxidation of methane. We suggest that significant upward migration of methane in the sedimentary column and its oxidation by sulfate may fuel the microbial activity in the sulfate methane transition zone (SMTZ). The biomass, created by this microbial activity, can be used by the iron reducers below, in the methanic zone of the sediments of the SE Mediterranean.

## 1 Introduction

Iron (Fe) is the fourth most abundant element in the Earth's crust. It appears as elemental Fe, Fe(II) and Fe(III), and has an important geobiological role in natural systems (e.g. Roden, 2006). Dissimilatory microbial iron reduction may be one of the first evolutionary metabolisms, and plays a key role in the reductive dissolution of Fe(III) minerals in the natural environment (Lovley and Phillips, 1986; Lovley

et al., 1987; Lovley and Phillips, 1988; Lovley, 1997; Weber et al., 2006) and in the mineralization of
organic matter in freshwater sediments (Roden and Wetzel, 2002). It also serves as a redox wheel that
drives the biogeochemical cycles of carbon, nitrogen, sulfur and phosphorous (Li et al., 2012 ; Slomp et
al., 2013; Sivan et al., 2014; Egger et al., 2016; Ettwig et al., 2016; Riedinger et al., 2017; März et al.,

45     2018).

Dissimilatory iron reduction is part of the anaerobic respiration cascade, in which different organic
substrates are used for energy by microorganisms and oxidized to dissolved inorganic carbon (DIC). This
is accomplished by reduction of electron acceptors, other than oxygen, according to their availability and
energy yield. Denitrification is the first respiratory process in anoxic sediments, followed by manganese
reduction, iron reduction and then sulfate reduction. Methane ($CH_4$) production (methanogenesis) by
archaeal methanogens is traditionally considered to be the terminal process of microbial organic matter
mineralization in anoxic environments, after the other electron acceptors have been exhausted (Froelich
et al., 1979). When the produced methane diffuses away from the methanic layer and meets an electron
acceptor it can be consumed by microbial oxidation (methanotrophy). In marine sediments anaerobic
oxidation of methane (AOM) coupled to sulfate reduction has been shown to occur (Iversen and
Jørgensen, 1985; Hoehler et al., 1994; Hinrichs et al., 1999; Boetius et al., 2000; Orphan et al., 2001;
Knittel and Boetius, 2009), and was found to consume up to 90 % of the methane that diffuses upward
to the sulfate methane transition zone (SMTZ) (e.g. Neiwöhner et al., 1998; Valentine, 2002).
The classical process of dissimilatory iron reduction is coupled to the oxidation of organic matter
(organoclastic iron reduction) (Eq. 1, Lovley, 1991; Lovley et al., 1996). However, iron reduction can be
coupled to other processes as well, such as hydrogen ($H_2$) oxidation (hydrogenotrophic iron reduction)
(Eq. 1, Lovley, 1991). Additionally, Fe(III) can be reduced microbially (and also abiotically) by pyrite
oxidation (Eq. 2, Bottrell et al., 2000), leading to sulfur (S) intermediates, and followed by their
disproportionation to sulfate and sulfide via a "cryptic" sulfur cycle (e.g. Holmkvist et al., 2011).
$$2Fe^{3+} + organic\ matter/H_2/humic\ acids \rightarrow 2Fe^{2+} + HCO_3^-/CO_2/2H^+ \qquad (1)$$
$$FeS_2 + 14Fe^{3+} + 8H_2O \rightarrow 15Fe^{2+} + 2SO_4^{2-} + 16H^+ \qquad (2)$$
Another recently discovered pathway of iron reduction is by AOM (Eq. 3).
$$CH_4 + 8Fe(OH)_3 + 15H^+ \rightarrow HCO_3^- + 8Fe^{2+} + 21H_2O \qquad (3)$$
This process in marine sediments was revealed using incubation experiments with marine seeps
sediments (Beal et al., 2009; Sivan et al., 2014). It was also suggested to exist in deep sea sediments
mainly through geochemical profiles and their modeling (Sivan et al., 2007; März et al., 2008; Riedinger
et al., 2014), and also in brackish coastal sediments (Slomp et al., 2013; Segarra et al., 2013; Egger et al.,
2014; Egger et al., 2016; Rooze et al., 2016;  Egger et al., 2017). In freshwater environments, it was
suggested to occur in lakes (Crowe et al. 2011; Sivan et al., 2011; Norði et al., 2013), and in denitrifying
cultures from sewage, where it was performed by methanogens (Ettwig et al., 2016). Iron-coupled AOM
in natural lake sediments was indicated using isotope pore-water depth profiles (Sivan et al., 2011), rate
modeling based on these profiles (Adler et al., 2011), microbial profiles (Bar-Or et al., 2015), and directly
from a set of sediment slurry incubation experiments (Bar-Or et al. 2017). The few microbial studies on
iron-coupled AOM (mainly in cultures) showed either the involvement of methanogenic/methanotrophic
archaea (Scheller et al., 2016; Ettwig et al., 2016; Rotaru and Thamdrup, 2016; Cai et al., 2018; Yan et
al., 2018) or a cooperation between methanotrophs and methanogens (Bar-Or et al., 2017).
Whereas Fe(II) is highly soluble, Fe(III) which is the most abundant species of iron under natural
conditions, appears as low-solubility oxidized minerals. This makes iron usage a challenge to
microorganisms, which need to respire these iron-oxide minerals, thus rendering many of the iron-oxide
minerals effectively unavailable for reduction and leading to the dominance of sulfate reducing bacteria
beyond a certain depth. Therefore, it is not trivial to observe iron reduction below the upper iron reduction
depth, in the methanic zone, where iron-oxides are assumed to be of low reactivity. Moreover, this type
of iron reduction is occasionally accompanied by depletion in methane concentrations, suggesting a
possible link between the iron and methane cycles. There are three potential mechanisms that can link
the cycles: 1) a competition between methanogens and iron-reducing bacteria over substrate, 2) a
metabolism switch of methanogens from methanogenesis to iron reduction, and 3) iron coupled AOM,
as mentioned above. Previous observations in other environments demonstrated the inhibition of
methanogenesis under iron-reducing conditions due to competition between methanogens and iron-
reducing bacteria for the common acetate and hydrogen substrates (Lovley and Phillips, 1986; Roden
and Wetzel, 1996; Conrad, 1999; Roden, 2003). Different methanogens can also utilize iron directly, by
reducing Fe(III). This was shown in pure cultures with the amorphous Fe(III) oxyhydroxide (Bond and
Lovley., 2002), in pure cultures close to natural sedimentary conditions (Sivan et al., 2016), in natural
lake sediments with different iron oxides (i.e. amorphous iron, goethite, hematite and magnetite) (Bar-or
et al., 2017), in anoxic ferruginous lake sediment enrichments (Bray et al., 2018), and in iron-rich clays
(Liu et al., 2011; Zhang et al., 2012; Zhang et al., 2013).
Despite the above studies, the nature of the link between the biogeochemical cycles of iron and methane
in the methanic zone of marine sediments, which creates suitable conditions for iron reduction, has not
yet been determined. Furthermore, this microbial iron reduction in methanic zones has not been shown
in the sediments of oligotrophic shallow marine environments. In this study we report the observation of
microbial iron reduction in the methanic depth of marine sediments from the oligotrophic SE
Mediterranean continental shelf. The microbial iron reduction is observed by using geochemical pore-
water profiles, qPCR profiles (of archaea, bacteria and the *mcrA* functional gene) and 16S rRNA gene
sequencing profiles at three different stations, combined with a simple slurry incubation experiment from
the methanic zone. The slurries were amended with hematite and magnetite. Given their low reactivity
these are the expected Fe(III) minerals to survive the sulfide zone (Canfield, 1989; Poulton et al., 2004).
Furthermore, these minerals were found to be active in iron-coupled AOM in lake sediments (Bar-Or et
al., 2017). The profiles, the incubation experiment as well as the related microorganisms, are discussed
in terms of the possible links between the cycles of iron and methane.

## 2 Methods

### 2.1 Study site

The surface water in the Levantine Basin of the SE Mediterranean Sea, including Israel's continental shelf, is an oligotrophic nutrient-poor marine system (Herut et al., 2000; Kress and Herut, 2001). The continental shelf narrows from south to north and is composed of Pliocene-Quaternary Nile-derived sediments. The sedimentation rate decreases with increasing distance from the Nile Delta and from the shoreline (Nir, 1984; Sandler and Herut, 2000). Off the shore of Israel, the sediment accumulation rate is relatively high at ~0.1 cm $y^{-1}$ (Bareket et al., 2016). The bottom seawater along the continental shelf is well oxygenated and sulfate concentrations at the water-sediment interface are ~30 mmol $L^{-1}$ (Sela-Adler et al., 2015). The central and eastern regions of the Levantine Basin have relatively low total organic carbon (TOC) content (~0.1 – 1.4%; Almogi-Labin et al., 2009; Sela-Adler et al., 2015; Astrahan et al., 2017) as compared to the Western Mediterranean Basin and offshore the Nile River delta (1 – 2%). Along the Egyptian coast, the TOC in surface sediments on the shelf reach maximum values of 1.5% (Aly Salem et al., 2013). The finding of a free gas zone, which is located from few to tens meters below the seafloor (i.e. gas front), in seismic profiles within the sediments of the continental shelf of Israel (Schattner et al., 2012), led to the discovery of biogenic methane formation at some locations in the shallow sediments (Sela-Adler et al., 2015).

### 2.2 Sampling

Seven sediment cores (~5 – 6 m long) were collected using a Benthos 2175 piston corer, from the undisturbed sediments of the SE Mediterranean continental shelf of Israel at water depths of 81 – 89 m from three stations; SG-1, PC-3 and PC-5 (Fig. 1). The cores were sampled during cruises of R.V. *Shikmona* between 2013 to 2017, and by the R.V. *Bat-Galim* on January 2017 (Table 1). The sediment cores were sliced on board every 25 – 35 cm within minutes upon retrieval from the seafloor. This area was previously investigated for other purposes, such as the sulfate reduction in the SMTZ (Antler et al., 2015; Wurgaft et al., 2019), and the possibility for methanogenesis (Sela-Adler et al., 2015).

From each interval, a 2.5 mL of sediment sample was collected and inserted immediately into an anoxic 10 mL glass bottle filled with 5 mL NaOH 1.5 N for headspace measurements of methane concentration (after Nüsslein et al., 2003). Approximately 3 mL of sediment was sampled every 50 cm for porosity. In addition, another 2.5 mL sediment sample was taken from each segment of the cores and transferred into a 20 mL glass bottle filled with NaCl saturated solution for $H_2$ concentrations measurements. Sediment samples from each segment of the cores were centrifuged on board if possible or in the lab within a day by Sorval centrifuge at 9299 g under 4 °C and Ar atmosphere in order to extract pore-water for chemical analysis. The supernatant was filtered (0.22 µm) and analyzed for Fe(II), sulfate, sulfide, DIC and the stable carbon isotope composition of the DIC ($\delta^{13}C_{DIC}$). After the pore-water extraction, the sediment was analyzed for the content of the different reactive iron minerals (Table 2). In addition, a sediment sub-sample from each segment of the January 2017 core from Station SG-1 was kept at -20 °C for molecular analysis. Due to high water content and movement in the uppermost part of the sediments, two ~30 cm sediment cores were also sub-sampled separately, using a 0.0625 $m^2$ box corer (Ocean Instruments BX

700 Al) and Perspex tubes during the September 2015 and January 2017 cruises. The short cores were
stored at 4 °C, cut in the lab within 24 hours after their collection and their results are presented for the
top sediment (Fig. 2a – d).

**2.3 Slurry incubation experiment**

The experimental set-up (Table 2) consisted of 11 bottles with sediment from the methanic zone (265-
285 cm depth) from Station SG-1, where iron reduction was apparent from the pore-water profiles (Fig.
2d). Prior to the beginning of the experiment, sediment from the designated depth had been homogenized
in an anoxic bag under $N_2$ atmosphere. It was then transferred under anoxic conditions to a 250 mL glass
bottle with the addition of synthetic sea water without sulfate to reach a 1:1 sediment:water slurry ratio
for a 3 months incubation period. After the incubation period the slurry was sub-divided anoxically to
the 11 experiment bottles (60 mL each), and synthetic sea water was added for final sediment:water ratio
of 1:3. The bottles were sealed with a crimped cap and were flushed with $N_2$ for 5 minutes, shaken
vigorously and flushed again, (repeated 3 times). Three experimental bottles were autoclaved twice to
serve as "killed" control for the experiment. The experimental bottles were amended with 1.6 g $L^{-1}$ of
hematite ($Fe_2O_3$) or 2.3 g $L^{-1}$ of magnetite ($Fe_3O_4$) to reach Fe(III) final concentration of 10 mmol $L^{-1}$.
The three killed bottles were amended with the iron oxides after they cooled down to room temperature.
$H_2$ was added to some treatments to test its potential as an electron donor. One mL of $H_2$ was injected by
gas tight syringe to the three killed bottles, to two bottles with the addition of hematite and to two bottles
with the addition of magnetite (to reach final concentration of ~4% of the head space volume). The
experimental bottles were sampled several times for dissolved Fe(II) concentrations during the 14 day
experiment period.

**2.4 Analytical methods**

**2.4.1 Pore-water analyses**
Methane concentrations in the pore-water were analyzed by Focus Gas Chromatograph (GC; Thermo)
equipped with FID detector with a detection limit of 50 µmol $L^{-1}$. To calculate the methane concentrations
the sediment porosity was considered. Porosity was determined by drying wet sediment samples at 60
°C until there was no weight loss (~48 h). It was calculated as the weight loss from the initial weight of
the samples. $H_2$ concentrations were analyzed in a Reducing Compound Photometer Gas Chromatograph
(RCP-GC; Peak Laboratories). Dissolved Fe(II) concentrations were measured using the ferrozine
method (Stookey, 1970) by a spectrophotometer at 562 nm wavelength with detection limit of 1 µmol $L^{-}$
$^1$. Sulfide was measured using the Cline (1969) method by a spectrophotometer at 665 nm wavelength
with detection limit of 1 µmol $L^{-1}$. Total sulfur concentrations were measured in an inductively coupled
plasma atomic emission spectrometer (ICP-AES), Perkin Elmer Optima 3300, with an analytical error of
±1% (average deviations from repeated measurements of a seawater standard). Since sulfide was not
detected in any of the sediment cores, the total sulfur concentration in each pore-water sample was
assumed to be the sulfate concentration of that sample. The $\delta^{13}C_{DIC}$ values were measured on a DeltaV
Advantage Thermo© isotope-ratio mass-spectrometer (IRMS) at a precision of ±0.1 ‰. Results are
reported versus VPDB standard. Pore-water profiles of dissolved total sulfur, $CH_4$, $\delta^{13}C_{DIC}$, Fe(II) and
$H_2$ were produced during the study, and all of them are presented (Fig. 2). For each profile where
duplicate samples were taken the error bar is that of the average deviation of the mean of the duplicates,
in cases where only single samples were taken, it is the analytical error (if larger than the symbol).
**2.4.2 Sediment analysis**
Reactive Fe(III) in the sediments was measured according to the Poulton and Canfield (2005) definition
and sequential extraction procedure. The different reactive iron minerals were separated to 1) carbonate-
associated Fe ($Fe_{carb}$) (i.e. siderite and ankerite); 2) easily reducible oxides ($Fe_{ox1}$) (i.e. ferrihydrite and
lepidocrocite); 3) reducible oxides ($Fe_{ox2}$) (i.e. hematite, goethite and akageneite) and 4) magnetite
($Fe_{mag}$). Sediment samples were dried at 60℃, then, approximately 0.6 g dry sediment was inserted to a
centrifuge tube with 10 ml of a specific extractant at every stage under atmospheric conditions and
constant agitation (Table 3). The fluids were separated from the sediment by centrifugation and removed
from the tube with Pasteur pipette after every extraction stage. At the end of each extraction stage, the
extractant was transferred to a 15 mL falcon tube with 0.1 mL ascorbic acid and 0.1 mL ferrozine solution
to reduce all the Fe(III) to Fe(II) and fix it, then it was measured spectrophotometrically. The results
presented as "total reactive Fe(III)" are the sum of $Fe_{ox1}$, $Fe_{ox2}$ and $Fe_{mag}$. The profile of pyrite ($Fe_{py}$) was
taken from Wurgaft et al. (2019).
**2.4.3 Quantitative PCR and 16S rRNA gene V4 amplicon pyrosequencing**
DNA was extracted from the sediment core of Station SG-1 from January 2017 using Power Soil DNA
Kit (MoBio Laboratories, Inc., Carlsbad, CA, USA) following manufacturer's instructions. Copy
numbers of selected genes were estimated with quantitative PCR (qPCR) as described previously (Niu
et al., 2017) using specific primers: Uni519f/Arc908R and bac341f/519r for archaeal and bacterial 16S
rRNA genes, respectively, and mlas/mcrA-rev for the *mcrA* gene, which encodes the α-subunit of methyl-
coenzyme M reductase. The amplification efficiency was 94.5%, 106.3% and 92.4% for the archaeal 16S
rRNA, bacterial 16S rRNA and the *mcrA* gene, respectively (the respective $R^2$ of the standard curve was
0.998, 0.998 and 0.995).
The V4 regions of bacterial and archaeal 16S rRNA genes were amplified using barcoded 515FB/806RB
primers (Walters et al., 2015) and Arch519/Arch806 primers (Song et al., 2013), respectively. PCR
mixture contained 6 – 10 ng total DNA, 5 μL 10× Ex Taq buffer, 4 μL 2.5 mmol $L^{-1}$ dNTP mix, 1 μL of
each primer, 0.25 μL Ex Taq polymerase (Ex-Taq; TaKaRa, Dalian, China) and 5 μL bovine serum
albumin (25 mg $mL^{-1}$) in a total volume of 50 μL. DNA was sequenced as 2x150 bp reads using Illumina
MiSeq platform (Illumina, USA). Sequence quality assessments, chimera detection and down-stream
phylogenetic analyses were conducted in QIIME (Caporaso et al., 2010). Taxonomic assignments for
each OTU were performed in QIIME using the BLAST method and the SILVA128 reference database.
24056 to 132042 high quality sequences were obtained per sample, with the proportion of high-quality
sequence versus total sequence between 81.97 – 99.89%. Spearman correlation was performed using the
online calculator (http://www.sthda.com/english/rsthda/correlation.php) to test the relevance of
microbial abundance and communities with Fe(II) concentration along the depth of the sediment core
from 185 cm to the bottom 575 cm, which is the methanic zone of the sediment core according to the
geochemical profile (see the results below).
**3 Results**
**3.1 Geochemical profiles**
Geochemical pore-water profiles of several sediment cores from the three stations (SG-1, PC-3 and PC-
5 (Fig. 1, Table 1)) were produced in order to characterize the iron reduction process in the methanic
zone of the SE Mediterranean continental shelf and to identify its potential sources. The pore-water
profiles at Station SG-1 (Fig. 2a) show complete depletion of total sulfur at approximately 150 cm depth
in all cores. Sulfide concentrations were below the detection limit in all cores, indicating that the total
sulfur is mostly sulfate. The methane concentrations in the pore-water (Fig. 2b) show an increase with
depth immediately after the consumption of sulfate. The maximum methane concentration was
approximately 10 mmol $L^{-1}$ at ~140 cm depth in June 2015. The other methane depth profiles show an
increase in the concentrations to approximately 2 mmol $L^{-1}$ and then leveling off throughout the bottom
of the cores (~600 cm). Detected dissolved Fe(II) concentrations (Fig. 2d) were found in the upper iron
reduction zone (between 30 – 90 cm depth), and a second peak was found in the deeper part of the
sediment, at the methanic zone (below 180 cm depth). Maximum dissolved Fe(II) concentrations reached
84 µmol $L^{-1}$ in the upper iron reduction zone of the sediments and 65 µmol $L^{-1}$ in the methanic zone. The
$\delta^{13}C_{DIC}$ values (Fig. 2c) were the lowest (-35 ‰) as expected at the SMTZ depth, and the highest in the
methanic zone. $H_2$ concentrations (Fig. 2e) decreased to a minimum of 0.017 µmol $L^{-1}$ at 155 cm depth,
and then increased to a maximum of 0.147 µmol $L^{-1}$ at 485 cm depth.
Pore-water profiles from Station PC-3 (Fig. 2g – l) show similar patterns to Station SG-1 on all three
sampling dates, but with lower methane concentrations. Total sulfur (Fig. 2g) was completely depleted
within the upper 300 cm depth. Sulfide concentrations were below the detection limit at this station as
well. Methane profiles show an increase in methane concentration immediately after the consumption of
sulfate. The maximum methane concentration (Fig. 2h) reached 0.8 mmol $L^{-1}$ at 450 cm depth in the
Aug-13 core. The dissolved Fe(II) profiles (Fig. 2j) show two peaks at this station as well, one in the
upper part of the sediment with maximum value of 32 µmol $L^{-1}$ at 177 cm depth, and another one with
maximum value of 64 µmol $L^{-1}$ at 390 cm depth at the methanic depth. The $\delta^{13}C_{DIC}$ values (Fig. 2i)
decreased from approximately -10 ‰ at the water-sediment interface to -20 ‰ at the SMTZ. Below that
zone there was an increase in $\delta^{13}C_{DIC}$ values to about -5 ‰ due to methanogenesis. $H_2$ concentrations
(Fig. 2k) remained around 2 µmol $L^{-1}$ along the core. The three deviating points that do not fit the clear
pattern are attributed to an analytical or sampling error.
Pore-water profiles from the core collected at Station PC-5 (Fig. S1) resemble the profiles of Station PC-
3. Total sulfur was depleted at approximately 300 cm, and methane concentrations increased below that
depth to 0.3 mmol $L^{-1}$. The Fe(II) profile shows two peaks in this core as well, one in the upper sediment
of 20 µM at 150 cm depth and the second of 30 µmol $L^{-1}$ in the methanic zone. The $\delta^{13}C_{DIC}$ value
decreased from -5 ‰ at the water-sediment interface to -25 ‰ at the SMTZ, and below that depth
increased to -17 ‰ at the methanic zone.
In addition to the dissolved constituents' profiles, reactive iron minerals were extracted from the sediment
collected on September 2015, and operationally defined iron mineral fractions profiles from Stations SG-
1 and PC-3 were produced (Fig. 2f and l). In Station SG-1 there appears to be a slight variability in the
content of the minerals (Fig. 2f). The $Fe_{carb}$ content in the upper part of the sediment was 0.22 dry wt%,
increased to ~0.45 dry wt % at 103 cm depth and then remained constant. The $Fe_{ox1}$ content was 0.49 dry
wt % in the upper part of the sediment, peaked at 203 cm depth to 0.64 dry wt % and then decreased to
0.50 dry wt % at the bottom of the core. The $Fe_{ox2}$ content was 2.15 dry wt % in the upper part of the
sediment, decreased to 1.03 dry wt % at 312 cm depth, and then it increased to 1.55 dry wt % at 427 cm
depth. $Fe_{mag}$ content was 0.34 dry wt % in the upper part of the sediment, decreased to 0.32 dry wt % at
153 cm depth, increased to 0.35 at 253 cm depth, decreased to 0.23 dry wt % at 312 cm depth, and
increased again to 0.35 dry wt % at the bottom. A pyrite content profile from Station SG-1 was also
produced (Wurgaft et al., 2019) from the September 2015 cruise and shows two peaks; the uppermost of
1.10 wt % at 153 cm depth, and the lower one of 1.80 wt % at 312 cm depth. The total reactive Fe(III)
oxides profile showed a general decrease from 3.00 dry wt % at 13 cm depth to 2.27 dry wt % at 507 cm
depth, with two minimum peaks of 2.42 dry wt % at 103 cm and of 1.88 dry wt % at 312 cm.
In Station PC-3 there appeared to be smaller changes in the different iron mineral fractions with depth
(Fig. 2l). The $Fe_{carb}$ content in the upper part of the sediment was 0.50 dry wt % and reached 0.69 dry wt
% in the deep sediment. The $Fe_{ox1}$ content was approximately 1.00 dry wt % throughout the sediment
column. The $Fe_{ox2}$ content was 0.78 dry wt % in the upper part of the sediment, increased to 0.89 dry wt
% at 167 cm depth and then decreased to 0.76 dry wt % at 495 cm depth. $Fe_{mag}$ content was 0.83 dry wt
% in the upper part of the sediment, increased to 0.89 dry wt % at 167 cm, and then decreased again to
0.75 dry wt % at 495 cm depth. The total reactive Fe(III) oxides content varied between 2.10 dry wt %
(at 167 cm depth) and 1.76 dry wt % (at 137 cm depth).
**3.2 Abundance and diversity of bacteria and archaea**
The qPCR of bacterial and archaeal 16S rRNA genes from the SG-1 core (collected on January 2017)
revealed an abundance of bacterial genes between $1.46 - 9.45 \times 10^6$ copies per g wet sediment, while that
of archaea was between $8.15 \times 10^5 - 2.25 \times 10^7$ copies per g wet sediment (Fig. 3). The abundance of
bacteria and archaea decreased gradually in the top 95 cm, increased sharply at 125 cm depth within the
SMTZ, remained relatively stable with high abundance at 185 – 245 cm (the top layer of the methanic
zone), and then decreased. Notably, the abundance of both bacteria and archaea peaked within the
methanic zone at 245 cm in correspondence with Fe(II) concentration peak. However, it is not feasible
to compare the abundance of archaea and bacteria by this method due to bias caused by the PCR primers
used (Buongiorno et al., 2017). The abundance of the *mcrA* gene (Fig. 3) increased sharply from the
surface layer to the SMTZ, peaked at 155 cm and remained stable at 155 – 245 cm, indicative of active
anaerobic methane metabolism in the SMTZ and an active methanic zone. Spearman correlation test
(Table S2) shows that the abundance of the bacteria and archaea 16S rRNA genes and *mcrA* genes
correlated with Fe(II) concentration in the methanic zone, where *mcrA* gene correlated the most
significantly ($r = 0.5429$, p value = 0.04789).
Illumina-sequencing of the 16S rRNA gene revealed diverse bacterial and archaeal communities
throughout the SG-1 core (Fig. 4). Although no clear plateau was observed on species rarefaction curve

for the current sequencing depth (Fig. S2), Shannon diversity indices reached stable values, indicating that those sequences well covered the diversity of bacterial and archaeal populations in the samples (Fig. S3). Shannon index, based on 16S rRNA gene sequences, shows higher diversity in the top layers of the sediment along with similar values through the core using the bacterial primers, while for sequences using archaeal primers, the values varied in different layers (Table S1). The bacterial sequences were affiliated with the following phyla: Planctomycetes (25.7%), Chloroflexi (23.2 %), Proteobacteria (12.9%), Deinococcus-Thermus (9.9 %), Acidobacteria (3.5%), Aminicenantes (3.3 %), Spirochaetes (2.3%), Deferribacteres (1.7%), Elusimicrobia (1.6%), Aerophobetes (1.6%), Nitrospirae (1.4%), Firmicutes (1.4 %), Actinobacteria (1.4 %), TM6 (Dependentiae) (1.2%), Marinimicrobia (SAR406 clade) (1.0%), and other taxa with less than 1% of the bacterial communities (Fig. 4a). Bathyarchaeota were the predominant archaea in all the sediment layers, based on the high relative abundance of their 16S rRNA gene sequences (91.0%). The remaining archaeal phyla comprised Euryarchaeota (3.2%), Thaumarchaeota (2.4%), Lokiarchaeota (1.0%), and other phyla with less than 1% of the archaeal communities (Fig. 4b). Spearman correlation analysis (Table S2) revealed that uncultured SBR1093 (r = 0.6176, p value = 0.01859) from bacterial Candidate Phylum SBR1093, subgroup 26 of Acidobacteria (r = 0.5841, p value = 0.02828), the uncultured bacterium from TK10 Class of Chloroflexi phylum (r = 0.5297, p value = 0.0544) and uncultured Bathyarchaeota sp. (archaea) (r = 0.5516, p value = 0.04388) correlated significantly with Fe(II) concentration.

**3.3 Incubation experiment**

Sediment from the observed deep iron reduction zone of Station SG-1 from January 2017 core was used for a simple short-term (couple of weeks) slurry incubation experiment in order to characterize the iron reduction process in the methanic zone. Hematite and magnetite, which were expected to survive the sulfate zone, and were shown to be a source for AOM in lake sediments, were added to the slurries. Indeed, the operationally defined iron mineral fractions profiles (Fig. 2f) confirm that hematite and magnetite were abundant in the methanic zone in this core.

The results of the experiment are shown in figure 5. Dissolved Fe(II) concentrations show significant increase from 11 $\mu$mol L$^{-1}$ to approximately 90 $\mu$mol L$^{-1}$ during the first three days in all the experimental bottles, except for the killed bottles, implying that the reduction is microbially mediated. Another observation was that the microorganisms were able to reduce both hematite and magnetite to the same extent. In addition, no difference in the Fe(II) concentrations between bottles with and without the addition of H$_2$ was observed.

**4 Discussion**

**4.1 General**

This study was performed in the SE Mediterranean (Fig. 1) above the area of a recently discovered gas front (Schattner et al., 2012). The investigated methane was found in the shallow sediments (~1-5 m deep) and seems biogenic based on its low $\delta^{13}C_{CH4}$ values and high C1/C2 ratio (Sela-Adler et al., 2015). Station SG-1 is located at the center of the gas front area, while Stations PC-3 and PC-5 are located at the edges, and indeed methane related processes were more intensive at Station SG-1. The source of this

gas front is not certain, but it was speculated to be terrestrial organic matter (Schattner et al., 2012). Our
results suggest that there are two sources for methane in the shallow sediment: the first is from migration
of methane from this gas front area (Wurgaft et al., 2019), and the second is from *in-situ* methane
formation, where the relative contribution of each source is currently unknown. *In-situ* methanogenesis
in the shallow shelf sediments is evident by the geochemical profiles of $\delta^{13}C_{DIC}$ and $\delta^{13}C_{CH4}$ (Sela-Adler
et al., 2015), by the microbial population abundance profile and by the functional *mcrA* gene profile
(Figs. 3 and 4, further discussed below). The TOC content in the methanic zone is ~0.8% at Station SG-
1 and ~1% at Station PC-3 (Sela-Adler et al., 2015), and these levels are known to be able to support *in-*
*situ* methanogenesis (Sivan et al., 2007).
The comparison between the sites show that methane reaches the highest concentrations at Station SG-1
(up to the saturation level (Sela-Adler et al., 2015)), specifically in the June 2015 profile (Fig. 2b). This
leads to intensive AOM by sulfate at the SMTZ, causing it to occur at shallower depth and to produce
lower $\delta^{13}C_{DIC}$ values than the other two stations. The relation between the upward fluxes of methane, the
SMT depth and the $\delta^{13}C_{DIC}$ values fit previous studies (e.g. Sivan et al., 2007). The higher methane
concentrations in the June 2015 profile is presumably due to intensive migration of methane from the
deeper sediments, and/or more intensive methane production at the exact location of the core collected
at that time. The $H_2$ concentrations at Station SG-1 (Fig. 2e) were lower by two orders of magnitude than
the concentrations at Station PC-3 (Fig. 2k), perhaps due to more intensive hydrogen consuming
processes at Station SG-1 (i.e. sulfate reduction, methanogenesis, iron reduction (Conrad et al., 1986;
Lovley, 1991). Dissolved Fe(II) pore-water profiles (Figs. 2d and j) show some variability between the
cores within the same station, probably as a result of environmental variations.
Despite the pore-water profiles variability between the stations, they show a resemblance in their trends.
All geochemical pore-water and iron mineral fraction profiles suggest that the sediments in this area of
the SE Mediterranean shelf can be classified into three general depth-zones (Fig. 2): **zone 1** is the upper
part of the sediment, where the classical iron reduction occurs, probably coupled to organic matter
oxidation, with sulfate reduction below it; **zone 2** is the SMT depth, where methane starts to increase,
sulfate is completely depleted, and Fe(II) (Fig. 2d and j) is either present in low concentrations or absent
(probably due to the precipitation of iron-sulfide minerals). In addition, the $\delta^{13}C_{DIC}$ values are the lowest
in this zone, as expected from the intensive AOM process there, which uses the isotopically light carbon
of the methane as a carbon source with small fractionation (Whiticar, 1999; Holler et al., 2009)**; zone 3**
is the methanic zone, where methane concentrations increased to the highest values in all stations, as did
the $\delta^{13}C_{DIC}$ since the carbon source for the methane comes mainly from $CO_2$, leaving the residual DIC
heavier by about 60 ‰ (Whiticar, 1999; Conrad, 2005). At this zone, local maxima of Fe(II)
concentrations in the pore-water were found in all cores, indicating reduction of iron oxides. The slurry
experiment results show only a slight increase in Fe(II) concentrations in the killed bottles compared to
their significant increase in the non-killed bottles, inferring that the iron reduction in zone 3 is microbial
(Fig. 5).

**4.2 Potential methanic iron reduction pathways**

This observed intensive iron reduction in the methanic sediments is the first discovered in the SE Mediterranean shelf. The phenomenon of iron reduction in the methanic depth has been observed before in other marine provinces (Jørgensen et al., 2004; März et al., 2008; Slomp et al., 2013; Riedinger et al., 2014; Treude et al., 2014; Oni et al., 2015; Egger et al., 2016). Yet, the type of link to the methane cycle is not well understood. Usually, iron reduction is coupled to oxidation of organic matter (Lovley and Phillips, 1988) and is performed by iron-reducing bacteria, which is probably the case in zone 1. It is however questionable if this also stands for zone 3 and if not, what process is responsible for the iron reduction at this depth and its relation to methane. The iron reduction in zone 3 can occur potentially via four pathways: 1) oxidation of organic matter arriving from the SMTZ, where it is produced by the microorganisms that live there and benefit from the upward migrating methane, 2) oxidation of the methane itself, 3) $H_2$ oxidation or 4) oxidation of sulfur species through a cryptic cycle.

The oligotrophic nature of the water column in the studied area would suggest that intensive bacterial iron reduction coupled with the oxidation of organic matter in zone 3 is less likely. Nevertheless, we observe high methane concentrations in zone 3 in all three stations, where part of it is from upward migration. This indicates that regardless of the surface water oligotrophic nature, the TOC substrate may be enough to sustain all the microbial activity and to take part in the iron reduction process in the methanic zone. This is possibly due to biomass production in the SMTZ (i.e. the microbial community including ANMEs and sulfate reducing bacteria (Boetius et al., 2000)) and its rapid use in the methanic zone (so the TOC content remains still low).

The importance of the methane flux as a carbon source that supports the deep microbial community at zone 2 and 3 in the sediments of the SE Mediterranean can be illustrated by comparing the organic carbon flux from the photic zone, with the flux of organic carbon that is oxidized by sulfate in the pore-water. Using traps, Moutin and Raimbault (2002) estimated an export flux of $7.4\pm6.3$ mgC m$^{-2}$ d$^{-1}$, which leaves the photic zone. However, Wurgaft et al. (2019) estimated that the flux of DIC toward the SMTZ from sulfate reduction is equivalent to $8\pm3$ mgC m$^{-2}$ d$^{-1}$. Whereas the difference between the two fluxes is statistically insignificant, it should be noted that the flux of organic material that survives aerobic oxidation in the water column and the upper part of the sediment column, as well as anaerobic oxidation by other electron acceptors with higher energy yield (Froelich et al., 1979; Emerson et al., 1980), is likely to be substantially smaller than the flux measured by Moutin and Raimbault (2002). Therefore, it is unlikely that export flux from the photic zone constitutes the sole source of carbon to the SMTZ. Wurgaft et al. (2019) suggested that "external" methane, originates in deeper portions of the sediments, provides an important source of carbon to the SMTZ in Station SG-1. Such fluxes of "external" methane are common along continental margin sediments (e.g. Milkov and Sassen, 2002; Milkov, 2004; Zhang and Lanoil, 2004; Paull et al., 2008; Fischer et al., 2013). Here, we suggest that this supply of methane, leads to intensive sulfate-mediated AOM in the SMTZ, and that this intensive process and biomass may serve as an additional substrate that "fuels" zone 3, activating the iron-oxides.

The recently discovered iron-coupled AOM process (Eq. 3) is the second potential process that can involve iron-oxide reduction in the deep methanic zone (Sivan et al., 2011: Segarra et al., 2013; Slomp

et al., 2013; Riedinger et al., 2014; Egger et al., 2015; Rooze et al., 2016; Egger et al., 2017; Bar-Or et
al., 2017). Fe(III) as an electron acceptor for AOM provides a greater free energy yield than sulfate
(Zehnder and Brock, 1980), and its global importance was emphasized (Sivan et al., 2011: Segarra et al.,
2013; Sivan et al., 2014). Two of the main environmental conditions for iron-coupled AOM to occur are
high dissolved methane concentrations and abundant reducible iron oxides (Riedinger et al., 2005;
Riedinger et al., 2014; Egger et al., 2017). Thus, from our profiles it seems that AOM could be a valid
option, considering the high methane concentrations and the high sedimentation rates (0.1 cm $y^{-1}$ (Bareket
et al., 2016)), which allow the iron oxides to survive the sulfidic zone and reach the methanic zone
(Riedinger et al., 2005; Riedinger et al., 2014; Egger et al., 2017). This can also be inferred from figure
6, where some association was observed between the dissolved Fe(II) concentrations and the methane
concentrations in zone 3. It seems that at high concentrations of Fe(II), methane concentrations are low
and vice versa. This could be a result of iron-coupled AOM that uses methane to reduce Fe(III)-oxides,
releasing dissolved Fe(II) to the pore-water. It can also suggest a type of competitive relationship between
methanogenesis and microbial iron reduction, or microbial population switching from methanogenesis
to iron reduction metabolism (e.g.  Sivan et al., 2016). It should be noted that our experiment was not
designed to test AOM due to its short time scale of a few weeks, hence another long experiment with the
addition of the $^{13}$C-labeled methane will be needed to shed more light on this association.
The third potential process that can be coupled to iron reduction in the methanic zone is $H_2$ oxidation. $H_2$
is an important intermediate in anoxic aquatic sediments. In this type of environment, it is produced
mainly by fermentation of organic matter (Chen et al., 2006), and can be involved in different microbial
processes where each process would need a certain amount of $H_2$ in order to occur (Lovley and Goodwin,
1988). The $H_2$ levels at Stations SG-1 and PC-3 (Fig. 2e and k) are relatively high in comparison to other
marine environments (Lilley et al., 1982; Novelli et al., 1987), suggesting that there is enough $H_2$ to
sustain the iron reduction process. The relatively high $H_2$ concentrations at these stations could be
explained by the dominance of $H_2$ production processes (i.e. fermentation (Chen et al., 2006)) compared
to $H_2$ consuming processes (i.e. sulfate reduction, methanogenesis, iron reduction (Conrad et al., 1986;
Lovley, 1991)). At Station PC-3, the $H_2$ concentrations (Fig. 2k) are constant in zone 3, this suggest that
in addition to being produced, $H_2$ is consumed as well. At Station SG-1 (Fig. 2e) there is a maximum
peak at zone 3, indicating that there is either more $H_2$ production or less $H_2$ consumption at this zone
compared to zone 2. This is reasonable considering the intensive microbial activity in zone 2. The
decrease in the $H_2$ concentrations below the peak suggests that $H_2$ consuming processes are intensive in
this zone. The $H_2$ involvement was tested by injecting 1 mL of this gas to the experimental bottles in the
methanic iron reduction process (Fig. 5). We observed that the increase of Fe(II) concentration was
similar in the bottles with $H_2$ addition compared to the bottles without $H_2$. This could mean that either
there is enough $H_2$ in the sediments as it is, as implied by the $H_2$ pore-water profiles, or that at the
methanic depth $H_2$ is not involved in the iron reduction process.
The fourth potential way to reduce iron in zone 3 is by an active sulfur cycle. The pyrite profile supports
this possibility by showing two peaks, uppermost in zone 2 of ~1 wt% and the other in zone 3 of ~2 wt%
at about 300 cm depth (Fig. 2f). The peak at 300 cm depth indicates possible active sulfur cycle, even

though sulfate is already undetected at 200 cm. Thus, a possible scenario is that Fe(III) is reduced by pyrite oxidation (Eq. 3) (Bottrell et al., 2000), which triggers the 'cryptic' sulfur cycle, as observed in other marine sediments (Holmkvist et al., 2011; Brunner et al., 2016; Egger et al., 2016). In this cycle, elemental sulfur, and eventually by disproportionation also sulfide and sulfate, are produced. The sulfide reacts with iron-oxide and precipitates as FeS or as pyrite ($FeS_2$) (Holmkvist et al., 2011). The sulfate can inhibit methanogenesis (Mountfort et al., 1980; Mountfort and Asher, 1981), which can result in the enhancement of the iron reduction process due to competition for substrate with the methanogenesis process. Another indication for an active sulfur cryptic cycle comes from the 16S rRNA sequencing analysis (Fig. 4), which shows that Proteobacteria, a potential sulfur related bacteria phylum, is one of the most abundant phyla in the sediments. Moreover, the increase in the abundance of Sva0485 order of the deltaproteobacteria class, a known sulfate reducer (Tan et al., 2019), with depth supports an active sulfur cycle in zone 3 as well.

**4.3 Potential microbial players**

Our data profiles and incubations indicate that the observed iron reduction in the methanic zone of the SE Mediterranean shelf is performed by microbial activity. The microbial results show first that the abundances of the bacteria and archaea (Fig. 4) are typical to oligotrophic marine sediments (e.g. South China Sea that contains ~0.5 – 1 % TOC (Yu et al., 2018)). Second, even though potential bacterial iron reducers, such as *Alicyclobacillus*, *Sulfobacillusin*, *Desulfotomaculum* genera (Firmicutes), *Acidiphilium* (Alphaproteobacteria), *Desulfobulbus*, *Desulfuromonas*, *Geobacter*, *Geothermobacter,* *Anaeromyxobacter* (Deltaproteobacteria) and *Shewanella* (Gammaproteobacteria) (Weber et al., 2006) comprise less than 0.1% of bacteria detected in the methanic zone (from 185 cm and below), it appears that both the microbial abundance and the Fe(II) concentration peaked at this zone. Cultivation efforts indicated that archaeal methanogens may also play a role in iron reduction within sediments (Sivan et al., 2016). Moreover, the relative abundance of methane-metabolizing archaea was shown to correlate with Fe(II) concentrations in Helgoland muds from the North Sea, where microbial abundance and the Fe(II) concentrations peaked at the methanic zone (Oni et al., 2015), similarly to the results found in the SE Mediterranean sediments. It is possible that methane-metabolizing archaea were involved in the iron reduction in the SE Mediterranean sediments, as the highest *mcrA* gene copies per gram wet sediment were detected in the SMTZ and in the top of the methanic zone (Fig. 3) where the Fe(II) concentrations are high (Fig. 2d). Methanotrophs, such as ANMEs, were found to be involved in iron-coupled AOM in marine and freshwater cultures (Scheller et al., 2016; McGlynn et al., 2015; Ettwig et al., 2016; Cai et al., 2018). ANMEs were found here with relatively low frequencies (ANME1, below 1% in most samples, circa 5% in the 185 cm layer), and their role in iron reduction within the SE Mediterranean sediments remains to be tested.

In our study, Spearman correlation analysis at Station SG-1 (Table S2) revealed that bacterial phyla SBR1093 (candidate Phylum), Acidobacteria and Chloroflexi, as well as archaeal Phylum Bathyarchaeota showed significant positive correlation with a Fe(II) concentration in the methanic zone. The Candidate Phylum SBR1093 was firstly identified in phosphate-removing activated sludge from a sequencing batch reactor (Bond et al., 1995), and is often detected in a short-chain fatty acid rich

environment such as wastewater treatment, and marine sediments (Wang et al., 2014). It was thought to be capable of growing autotrophically, but the metabolic capabilities related to iron reduction remain unclear. Strains of Acidobacteria and Chloroflexi phylum were found to be capable of iron reduction (Kawaichi et al., 2013; Kulichevskaya et al., 2014). In addition, members of Acidobacteria were found in iron-coupled AOM enrichment (Beal et al., 2009). The metabolic properties of Subgroup 26 from Acidobacteria and TK10 Class of Chloroflexi are still not known. Bathyarchaeota are globally distributed and account for a considerable fraction of the archaeal communities in the marine sediments, particularly, in the Mediterranean Pleistocene sapropels (Coolen et al., 2002; Zhou et al., 2018). While Bathyarchaeota have diverse metabolic capabilities (Lloyd et al., 2013; Meng et al., 2014; Evans et al., 2015; He et al., 2016; Yu et al., 2018; Feng et al., 2019), their role in iron reduction warrants further studies, as suggested from their high abundance here. Therefore, iron reduction and methane cycling within the deep methanic zone may be facilitated by an interplay among bacterial and archaeal groups, whose physiology and functions needs further investigation.

**5 Conclusions**

Our study used combined geochemical and microbial profiles together with a slurry incubation experiment to show microbial iron reduction in methanic sediments, and the potential microbial population performing this reduction. The Spearman analysis points out several potential microbial players (both bacterial and archaeal) that correlate to the dissolved Fe(II) profiles (e.g. Bathyarchaeota, Acidobacteria and Chloroflexi). Moreover, our study emphasizes that this iron reduction in the methanic zone can occur even in sediments of oligotrophic seas such as the SE Mediterranean. We suggest that the availability of iron minerals for reduction is linked to intensive upward fluxes of methane and high sulfate-AOM rates that may produce available biomass or/and hydrogen, which fuel deeper microbial processes. The deep iron reduction may also be linked to a cryptic sulfur cycle and iron-coupled AOM.

**5 Author contribution**

H.V and O.S designed research; B.H and O.S. were the PIs of the cruises; H.V, E.W and L.L performed research and analyzed the data; H.V, O.S, B.H, F.W, M.RB and L.L synthesized the data and wrote the paper.

The authors declare that they have no conflict of interest.

**6 Acknowledgments**

We thank the captain and crew of the R/V Shikmona and R/V Bat Galim from the Israel Oceanographic and Limnological Research for all their help during field sampling. Many thanks to E. Eliani-Russak for her technical assistance in the lab and to V. Boyko for her help with the reactive iron speciation procedure. We also thank all of Prof. O. Sivan's lab members for their help. We would like to thank also to the anonymous reviewers for their helpful and constructive comments. This study was supported by the joint grant of Israel Science Foundation and the National Natural Science Foundation of China (ISF-NSFC) [grant number 31661143022 (FW) and 2561/16 (OS)]. Funding was provided to H. Vigderovich by the Mediterranean Sea Research Center of Israel.

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

**Table 1**: Cores sampling details: dates, water depths and locations.

| Date | station | water depth (m) | Latitude | Longtitude |
|---|---|---|---|---|
| **August 14, 2013** | PC-5 | 87 | 32°55.47' | 34°54.01' |
| | PC-3 | 81 | 32°55.29' | 34°54.14' |
| **February 6, 2014** | PC-3 | 82 | 32°55.30' | 34°54.14' |
| **January, 2015** | PC-3 | 82 | 32°55.30' | 34°54.14' |
| **June 9, 2015** | SG-1 | 89 | 32°57.87' | 34°55.30' |
| **September 17, 2015** | SG-1 | 84 | 32°57.91' | 34°55.27' |
| **January 24, 2017** | SG-1 | 85 | 32°57.51' | 34°55.15' |


**Table 2**: Experimental set-up of the slurry incubation experiment.

| Treatment | Number of bottles |
|---|---|
| **Hematite** | 2 |
| **Magnetite** | 2 |
| **Hematite + H$_2$** | 2 |
| **Magnetite + H$_2$** | 3 |
| **Killed + hematite + H$_2$** | 2 |
| **Killed + magnetite + H$_2$** | 1 |


**Table 3**: Summary of reactive iron extraction procedure (after Poulton and Canfield, 2005).

| Extractant | Target compounds | Analyzed species | Formula | Shaking time (h) |
|---|---|---|---|---|
| **Magnesium chloride** | Ion-exchangeble Fe(II) | Adsorbed ferrous iron | $Fe^{2+}$ | 2 |
| **Sodium acetate** | Iron carbonates | Siderite Ankerite | $FeCO_3$ $Ca(Fe^{+2},Mg^{+2},Mn^{+2})(CO_3)_2$ | 24 |
| **Hydroxylamine hydrochloride** | "Easily reducible" Iron(hydr)oxides | Ferrihydrite, Lepidicrocite | $Fe^{3+}_2O_3*0.5(H_2O)$ $\gamma$-FeOOH | 48 |
| **Sodium dithionite** | "Reducible" oxides | Goethite, Hematite, Akageneite | $\alpha$-FeOOH $Fe_2O_3$ $\beta$-FeOOH | 2 |
| **Ammonium oxalate** | Poorly crystalline | Magnetite | $Fe_3O_4$ | 6 |


 **Figures captions:**

**Figure 1**: A map of the study area with the location of the three stations that were sampled SG-1, PC-3 and PC-5 (after Wurgaft et al., 2019).

**Figure 2**: Geochemical pore-water profiles of: total S, $CH_4$, $\delta^{13}C_{DIC}$, dissolved Fe(II), $H_2$ and extractable Fe fractions from sediment cores collected at the two stations: SG-1 (a-f) and PC-3 (g-l) in the SE Mediterranean. The profiles are divided roughly into three zones according to the dominant processes: upper microbial iron and sulfate reduction, sulfate-methane transition zone (SMTZ), and the methanic zone at the deep part. The dashed line in the $CH_4$ graph at SG-1 station represents the $CH_4$ saturation value in the pore-water (Sela-Adler et al., 2015). The following extractable Fe fraction profiles of stations SG-1 (f) and PC-3 (l) are from the September 2015 and January 2015 cruise (respectively): $Fe_{carb}$ (●), $Fe_{ox1}$ (■) $Feo_{x2}$ (▲), $Fe_{mag}$ (▼), $Fe_{py}$ (◆) (Wurgaft et al., 2019) and total reactive iron (●). The error bars for $CH_4$ are presented where duplicate sediment samples were collected. The error bars for Fe(II), $\delta^{13}C_{DIC}$ and $H_2$ are presented where measurements from the same sample were repeated at least twice. The analytical errors were too small to be displayed.

**Figure 3**: Sedimentary depth profiles of bacterial and archaeal 16S rRNA and *mcrA* functional genes of station SG-1 from January 2017, divided to three zones (as described in figure 2). Triplicates were produced from each sample with error bars smaller than the symbols displayed.

**Figure 4**: Phyla level classification of bacterial (a) and archaeal (b) diversity in the sediments of Station SG-1 from January 2017.

**Figure 5**: Dissolved Fe(II) results of the sediment slurry incubation experiment. The sediment was collected from Station SG-1 on January 2017 from sediment depth of 265-285 cm. The error bars were smaller than the symbols displayed.

**Figure 6**: The relationship between dissolved Fe(II) concentrations and methane concentrations in zone 3 of (a) Station SG-1 and (b) Station PC-3. An inverse association is observed between the two species, suggesting a relationship of competition or iron-coupled AOM.

**Figures**:
**Figure 1**

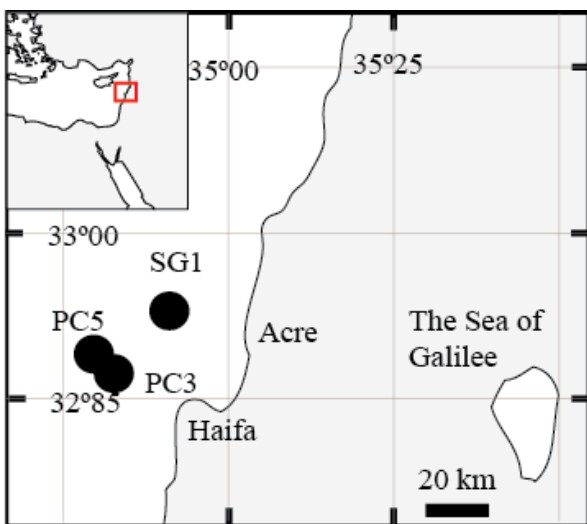






























**Figure 2**






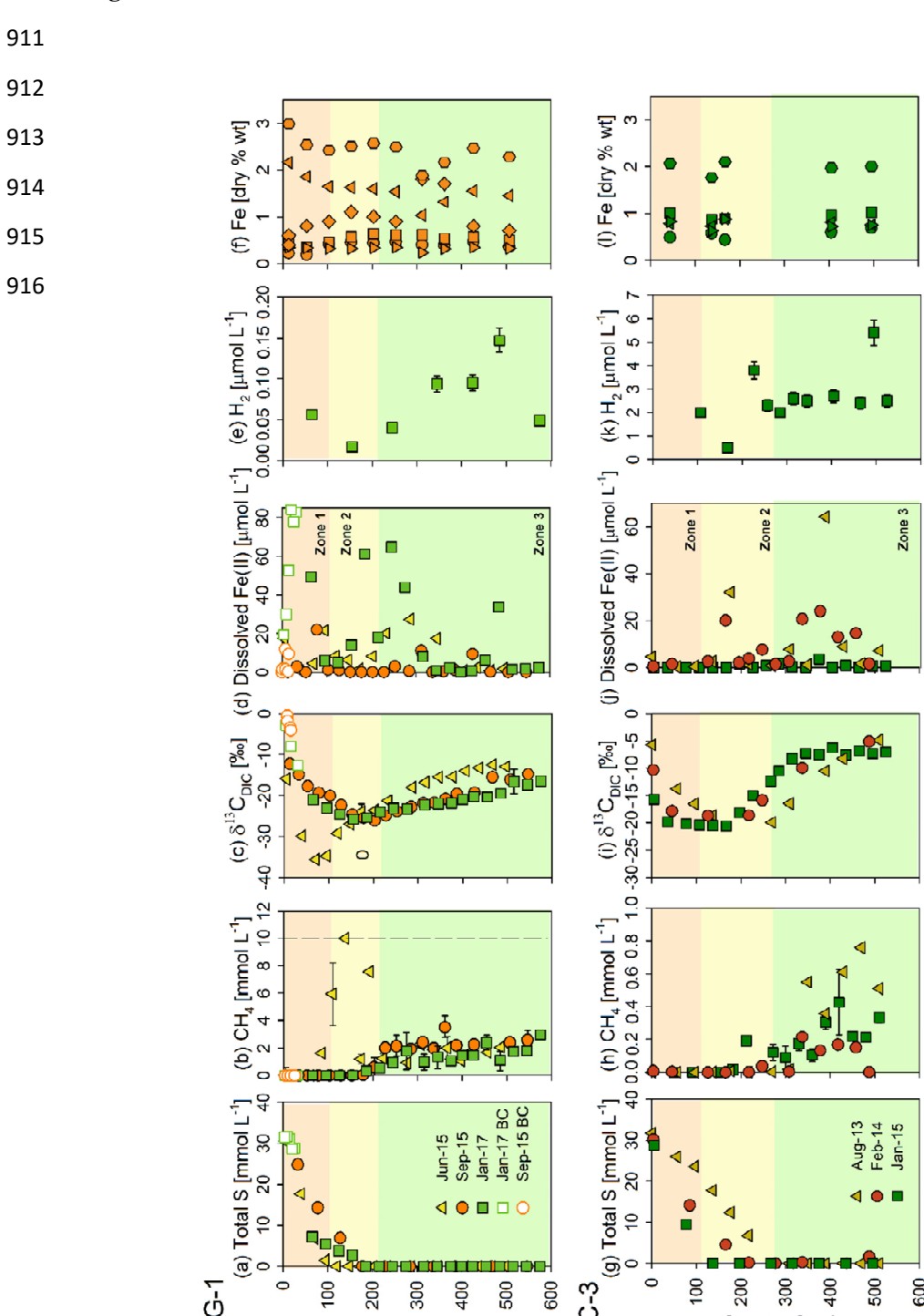

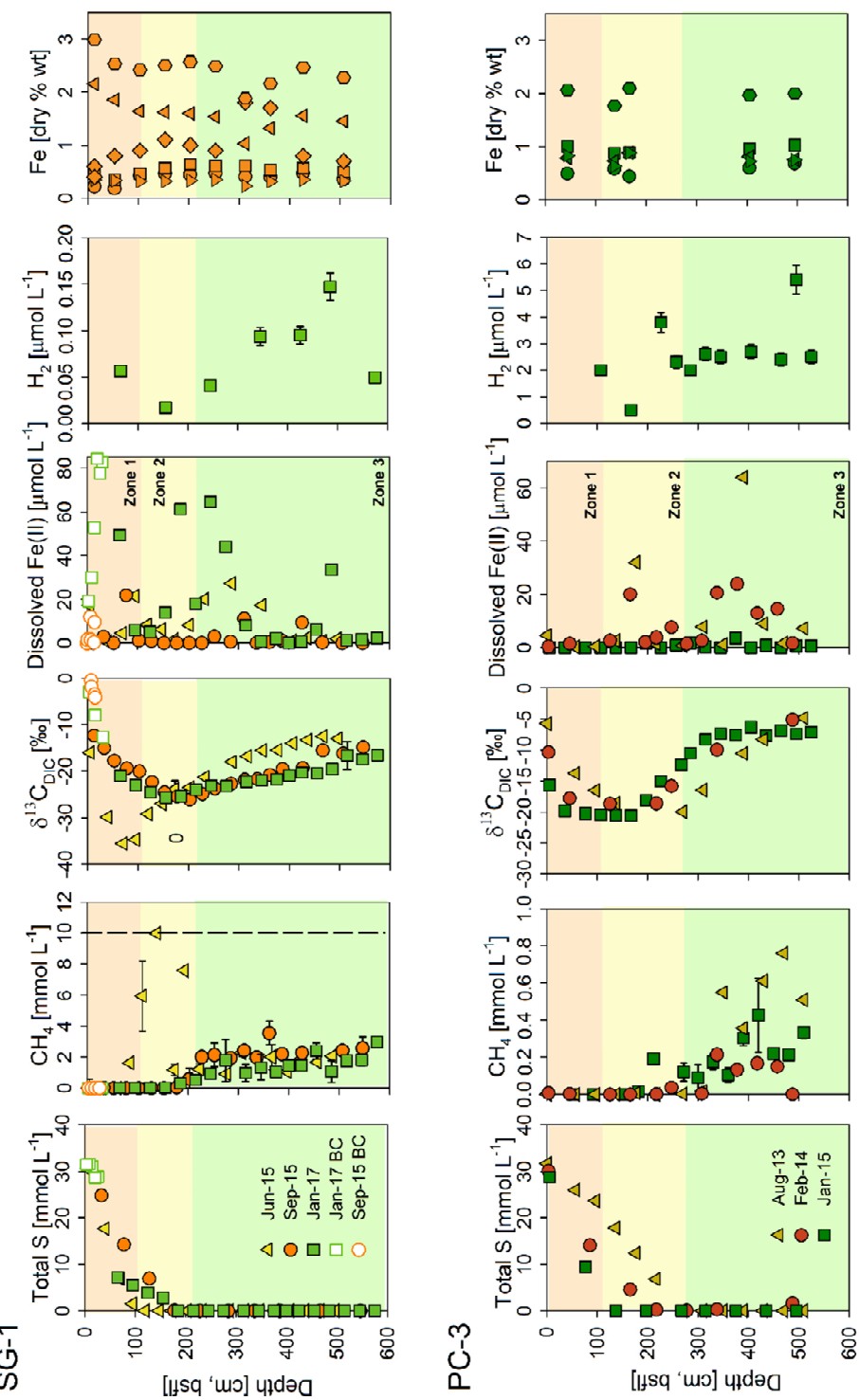

**Figure 3**




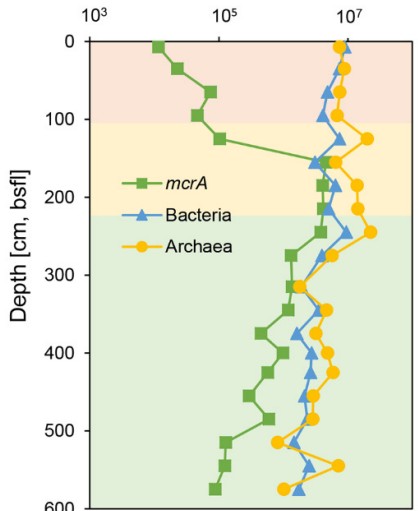

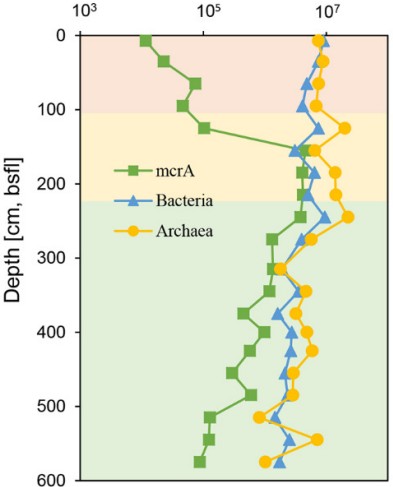

**Figure 4**



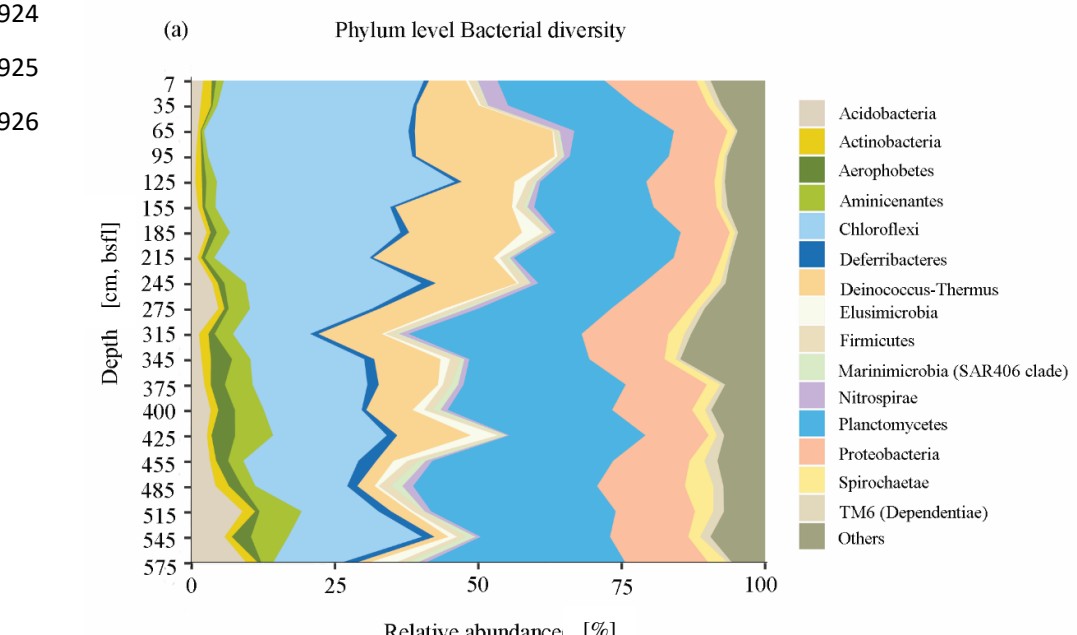

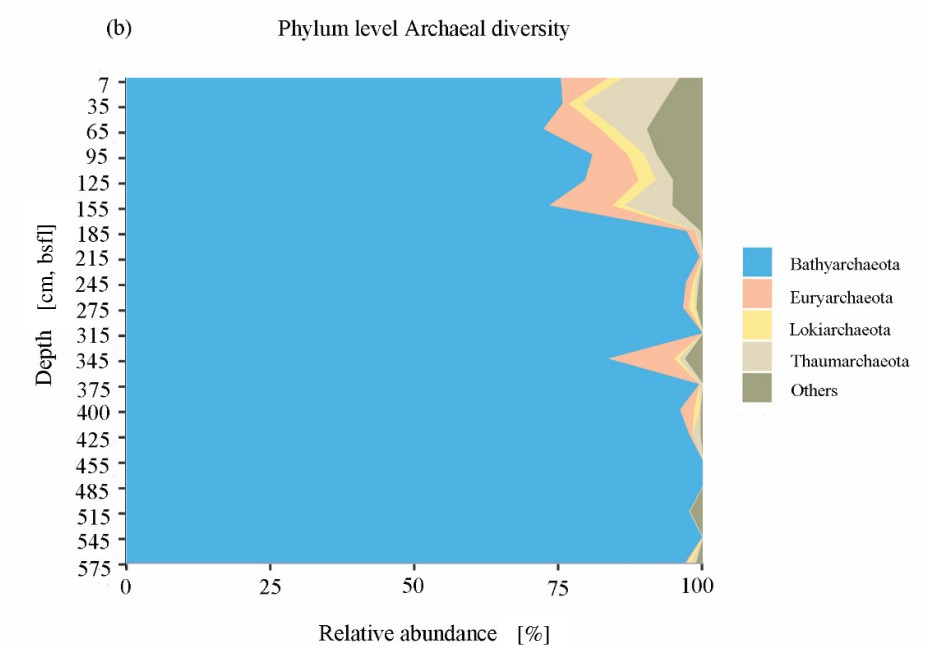

**Figure 5**

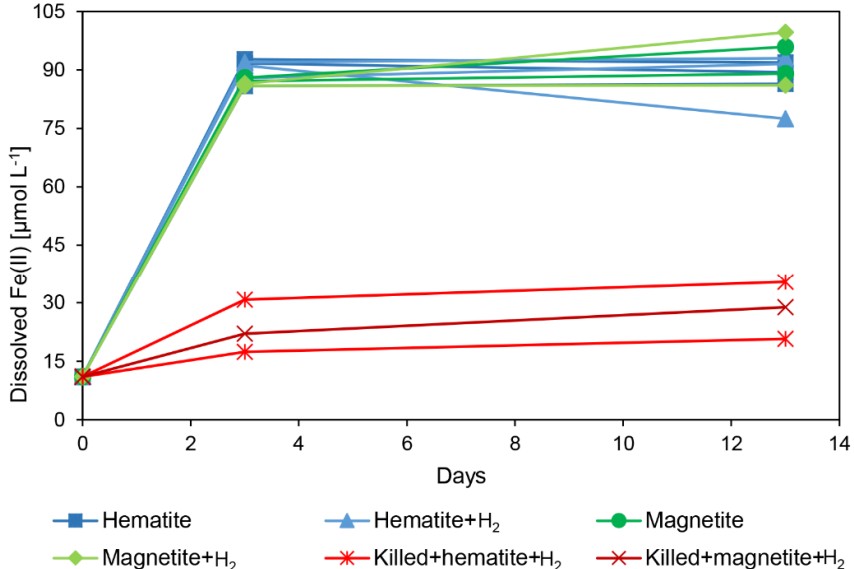



**Figure 6**

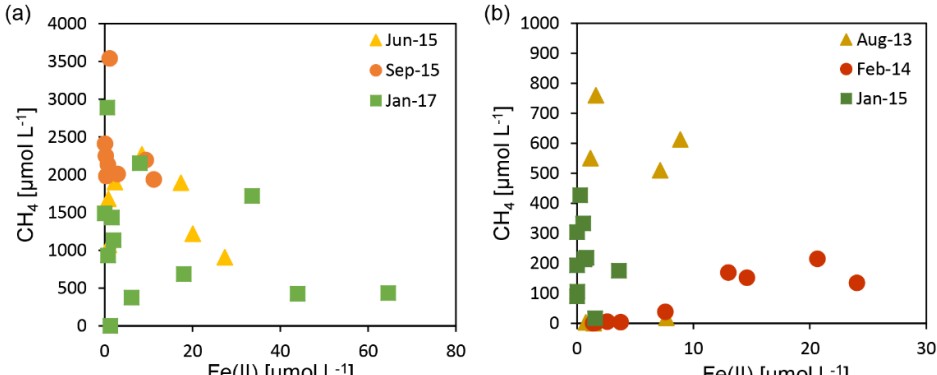







