# Peer review of "Evidence for microbial iron reduction in the methanic"

_Biogeosciences, 2019_

## Referee Comment (RC1) · Anonymous Referee #1 · 9 Feb 2019

General comments: This manuscript reports the first evidence for microbial iron reduction in a methanogenic zone in an oligotrophic shallow marine environment. The subject matter is within the scope of BG. The title is appropriate, the abstract is concise and complete, and the paper is well structured. Below I highlight comments in order to further strengthen the manuscript: 1) More information is needed about the rationale for the incubation experimental design. There is no mention in the methods of how the killed-control incubations were "killed". Was Fe added to the incubations? Was this before or after "killing"? How are the authors certain that the killing mechanism did not change the Fe(III) mineralogy? Please also clarify why hematite and magnetite were added instead of the more commonly used ferrihydrite and goethite; would these forms

of iron oxides be more environmentally relevant in the methanogenic zone? Please state explicitly with citations if that is the rationale. Please provide citations for hydrogen utilization as an electron donor for hematite and magnetite reduction to justify this experiment. 2) The data for the Fe extracts seem to be missing. The methods state (lines 171-172): "The different reactive iron oxides were separated to (1) carbonate associated Fe; (2) easily reducible oxides; (3) reducible oxides and (4) magnetite." Yet the only data for iron speciation has all four of these clumped together. It would be much more informative to know the depth profiles of the four individual species instead of the sum. Why wasn't this reported?

Specific comments: Line 86-90: also shown in anoxic ferruginous lake sediment enrichments, Bray et al 2018 Geobiology "Shifting microbial communities sustain multi-year iron reduction and methanogenesis in ferruginous sediment incubation" Line 248: therefore \*are\* not discussed... Line 333: regardless \*of\* the area's... Line 342: "The $H_2$ levels at stations SG-1 and PC-3 (Fig. 1) are relatively high (Lilley et al., 1982; Novelli et al., 1987), suggesting that there is enough $H_2$ to sustain the iron reduction process." Need to clarify that $H_2$ is two orders of magnitude higher at SG-1 vs. PG-3 and implications of that finding in relation to other findings. More text is needed explaining the references. Are these cited to show that $H_2$ has been historically high at the site? Need a reference for the second half of the statement about the $H_2$ concentrations needed for reduction of these Fe(III) minerals. Line 351-357: any evidence for or against this hypothesis in this study? Line 396: positive or negative correlation? Need to cite that supp material where this data is shown. Figure 1: Add zeroes before the decimal place on the x-axis. Increase the interval if it doesn't fit as is. Figure 2 and Figure S1: Move these to be included in Fig 1. Would enable easier comparison to have all the data together and will fit on a full-page graph oriented horizontally on page. Please add zones 1, 2, 3 like in Fig 1 to data in Figure 2 and Figure S1. Could move PG-5 depth profiles to supplementary. Figure 3: This color scheme and stretched out horizontal lines with white gaps in between is extremely hard to look at. Please fix to make it easier to see. Figure 5: I don't see a clear inverse relationship like the figure

caption states. I think you can say that there is more variability in methane at low Fe(II) and that methane is low for the few data points at the highest Fe(II).

---

## Author Comment (AC1) · 27 Feb 2019

Response to comments from anonymous reviewer #1 (our response is below each comment)

This manuscript reports the first evidence for microbial iron reduction in a methanogenic zone in an oligotrophic shallow marine environment. The subject matter is within the scope of BG. The title is appropriate, the abstract is concise and complete, and the paper is well structured. Below I highlight comments in order to further strengthen the manuscript.

[Figure]

We thank the reviewer for the positive and constructive review. We addressed and accepted almost all comments (see below) and revised the manuscript accordingly.

1. More information is needed about the rationale for the incubation experimental design. There is no mention in the methods of how the killed-control incubations were "killed". Was Fe added to the incubations? Was this before or after "killing"? How are the authors certain that the killing mechanism did not change the Fe(III) mineralogy? Please also clarify why hematite and magnetite were added instead of the more commonly used ferrihydrite and goethite; would these forms be more environmentally relevant in the methanogenic zone? Please state explicitly with citations if that is the rationale. Please provide citations for hydrogen utilization as an electron donor for hematite and magnetite reduction to justify this experiment.

We agree with the reviewer that additional details regarding the incubation experimental design were needed. The revised manuscript includes the specific information regarding the "killed" bottles (sediment killing via autoclave) and the addition of the hematite/magnetite to those bottles (after autoclaving). We believe the mineralogy of the Fe(III) that we added was not changed in the killed control since we added it after the autoclave stage when the killed bottles were already cooled down. The rational of adding hematite and magnetite is: a. These minerals are expected to survive the sulfate zone (amorphous iron or goethite are expected to be used first) and to remain in the methanogenic zone. These minerals were indeed found in our methanogenic sediments (see the supplementary figure). b. These were the minerals that were reactive in Fe-AOM in our previous work in lake sediments (Bar-Or et al., 2017) and seeps (Sivan et al., 2014). c. Magnetite is a conductive mineral, and has been shown to be preferred in some microbial processes due to this characteristic (e.g. Cruz et al., 2014; Tang et al., 2016; Rotaru et al., 2018). d. As these were the tested minerals, we added also $H_2$, a known electron donor for iron reduction, to these experiments. $H_2$ was shown also to reduce magnetite by S. putrefaciens (Kostka and Nealson, 1995). All of the above is explained and clarified in the revised version.

[Figure]

2. The data for the Fe extracts seem to be missing. The methods state (lines 171-172): "The different reactive iron oxides were separated to (1) carbonate associated Fe; (2) easily reducible oxides; (3) reducible oxides and (4) magnetite." Yet the only data for iron speciation has all four of these clumped together. It would be much more informative to know the depth profiles of the four individual species instead of the sum. Why wasn't this reported?

We decided it would be more relevant to show the whole amount of reactive Fe(III) in the sediments in order to show generally how much reactive Fe(III) minerals are in our sediments, considering that all four types of oxides can act as a potential electron acceptor. Nevertheless, we agree that it would be more informative to show the depth profiles of the four separately. Thus, In the revised manuscript we present a different figure containing all four types of Fe oxides.

Specific comments:

1. Line 86-90: also shown in anoxic ferruginous lake sediment enrichments, Bray et al 2018 Geobiology "Shifting microbial communities sustain multiyear iron reduction and methanogenesis in ferruginous sediment incubation"

We thank the reviewer for this comment, and we will add the citation (Bray et al., 2016).

2. Line 248: therefore *are* not discussed. . . Line 333: regardless *of* the area's.

We accept the reviewer's correction; and the revised manuscript is corrected.

3. Line 342: "The H2 levels at stations SG-1 and PC-3 (Fig. 1) are relatively high (Lilley et al., 1982; Novelli et al., 1987), suggesting that there is enough H2 to sustain the iron reduction process." Need to clarify that H2 is two orders of magnitude higher at SG-1 vs. PG-3 and implications of that finding in relation to other findings. More text is needed explaining the references. Are these cited to show that H2 has been historically high at the site? Need a reference for the second half of the statement about the H2 concentrations needed for reduction of these Fe(III) minerals.
We thank the reviewer for pointing out that this paragraph needs elaboration. We believe that the H2 concentrations at PC-3 station is higher than at SG-1station because the gas front is shallower at SG-1, which causes metabolic processes to be more intense and less amounts of H2 can accumulate in the sediments. The text is revised to clarify this subject. The cited papers are there to show that the known H2 concentrations in marine environments are lower than what we found in both stations. Reference for the second part of the sentence was added.

4. Line 351-357: any evidence for or against this hypothesis in this study?

Cryptic sulfur cycle is observed more and more in marine sediments (e.g. Holmkvist et al., 2011; Brunner et al., 2016). It seems that this cycle is possible here based on the microbial populations that contain those that may be involved in sulfur cycling (from 16S analysis). Also, pyrite was found in the methanogenic zone. We add and clarify this point in the revised version.

5. Line 396: positive or negative correlation? Need to cite that supp material where this data is shown.

We agree with the reviewer, the type of correlation and the cite to the supplementary material should have been mentioned. It is a positive correlation and is mentioned now in the revised manuscript.

6. Figure 1: Add zeroes before the decimal place on the x-axis. Increase the interval if it doesn't fit as is.

We accept the reviewer's comment, and the figure in the revised manuscript is adjusted accordingly.

7. Figure 2 and Figure S1: Move these to be included in Fig 1. Would enable easier comparison to have all the data together and will fit on a full-page graph oriented horizontally on page. Please add zones 1, 2, 3 like in Fig 1 to data in Figure 2 and Figure S1. Could move PG-5 depth profiles to supplementary.

We accept the reviewer's comment, and the figure in the revised manuscript is adjusted accordingly.

8. Figure 3: This color scheme and stretched out horizontal lines with white gaps in between is extremely hard to look at. Please fix to make it easier to see.

We accept the reviewer's comment, and the figure in the revised manuscript is adjusted accordingly.

9. Figure 5: I don't see a clear inverse relationship like the figure caption states. I think you can say that there is more variability in methane at low Fe(II) and that methane is low for the few data points at the highest Fe(II).

We agree with this comment and change the term "inverse correlation" to this more accurate sentence in the revised version.

---

## Referee Comment (RC2) · Anonymous Referee #2 · 13 Mar 2019

This manuscript presents pore-water data (S, CH4, Fe2+, H2, d13C-DIC), results of incubation experiments as well as data on the abundance and diversity of bacteria and archaea in sediments of the South Eastern Mediterranean continental shelf. Besides a typical zone of organoclastic iron reduction observed close to the sediment surface the authors report a second zone of enhanced Fe2+ pore-water concentrations within the methanic sediments below the sulfate/methane transition. Evidence for iron reduction in methanic subsurface sediments is commonly found in high accumulation continental shelf and margin sediments and a strong research interest currently exists in elucidating which (bio)geochemical pathways and potential microbial organisms mediate this "deep" iron reduction.

[Figure]

In this respect, the paper focusses on an important and topical research question and is in principle suitable for Biogeosciences. However, I regret to say that the manuscript has numerous flaws and appears as if it has not been prepared with the required care. The manuscript thus needs a major overhaul before I can recommend publication. The English also requires quite some polishing and I would suggest to ask an English native speaker to proofread the manuscript. There are numerous typos (which I have not all corrected in detail) and the wording is imprecise in many places – all this needs careful checking and correction.

Several issues that need to be considered when preparing a revised version:

1) The most important point is that the discussion is not adequate as it stands, several assumptions are not supported by the data and many key publications have not been cited. Often statements occur in the form of single sentences without "really" discussing the data obtained in the framework this study.

2) It is not clear to me which novel findings your study contributes to the topic of deep iron reduction. This needs to be outlined precisely.

3) Please provide a map that shows the study area and the three sampling locations and a table that summarizes the dates, exact positions, precise names etc. of the samples used in this study.

4) Please, also add a table that gives the details of the sequential extractions performed in this study.

5) Referencing is not adequate – i.e. several relevant papers are missing. I have listed some publications below but a careful literature search should be performed.

6) Please, precisely distinguish between and separate Results and Discussion. The Results chapter already contains a lot of interpretation/discussion and several references, which is formally incorrect.

Specific comments
Line 2 and throughout the manuscript: I do not like the term "methanogenic" very much because it implies that methane formation occurs in the respective sediment layer/interval. Based on your considerations on page 3 (lines 102 ff. and lines 112 ff.) concerning the current oligotrophic conditions in the study area as well as the deeper gas front detected based on seismic profiling, I suggest that it is likely that methane is diffusing/migrating up from deeper layers into the sediment depths investigated in this study. I would thus propose to speak of "methanic" sediments, which is more neutral.

L. 20: What exactly do you mean by "mechanistic" nature?

L. 25: delete "cores"; in the deeper methanic zone

L. 27: Do you mean $Fe_{2+}$ concentrations in pore water?

L. 37: Li et al. (2012) is only one of a vast amount of literature on this topic – you my add a few other papers. So change to (e.g. Li et al., 2012; Riedinger et al., 2017 (Frontiers in Earth Science); März et al., 2018 (Mar. Geol.).

L. 45: What exactly do you mean with "outward" diffusing methane? This is not clear to me. Please, specify.

L. 47: Key papers on sulfate-mediated AOM are missing here: please add at least Hinrichs et al. (1999) and Boetius et al. (2000). . . . it should then read: (e.g. Hoehler et al., 1994; Hinrichs et al., 1999; Boetius et al., 2000) . . .. and you may of course add further papers.

L. 49: Also here Valentine (2002) is only one example of a vast amount of literature on this topic. You may also wish to cite Niewöhner et al. (1998), GCA, here.

L. 51: Has to be iron "reduction" (instead of oxidation)

Ls. 58/59: Please, give the respective references.

L. 60: Please rephrase to: . . . incubation of marine seep sediment . . ..

Ls. 61 ff.: Please also cite the following papers in this context: März et al. (2008), Oni et al. (2015), Egger et al. (2018), who have also presented evidence for Fe-coupled AOM in marine, coastal, and brackish sediments.

Ls. 68 ff.: Please, also cite Oni et al. (2015) here who have presented microbial studies for the methanic zone of North Sea sediments.

Ls. 74 ff.: This sentence is hard to follow and sounds a bit odd. Please, rephrase.

L. 79: I would not speak of "inactive" in this context but rather of "of low reactivity". Furthermore, I do not find it surprising that reactive iron oxides are preserved and present below the SMT. This finding has already been explained by several studies/papers – amongst others by Riedinger et al. (2005), GCA, März et al. (2008), Mar. Geol., and März et al. (2018), Mar. Geol.

L. 87 ff.: You may also wish to cite Oni et al. (2015) here.

L. 92: What exactly do you mean with "reactivate" in this context? This is not clear to me – please specify. Were the Fe oxides "unreactive" before? By which process/condition have they been "reactivated"?

L. 97: What precisely is a "basic" incubation experiment?

L. 99: Please, rephrase to: . . . possible links between the cycling of iron and methane".

L. 102: I find it hard to imagine that the Levantine Basin is really one of the most oligotrophic marine settings in the world. I thought that globally the most oligotrophic ocean area is the South Pacific Gyre?! Please, check carefully and rephrase accordingly.

L. 109: I do not believe that the TOC contents are/were really "zero". I think this is an issue of the detection limit of the specific analytical method used. Please check.

Ls. 111 ff.: I do not understand the argumentation in this sentence. How can you conclude that methane found in shallow sediments is of biogenic origin if a deep gas front has been detected by seismics? Are you sure that the methane found in the shallo

sediments investigated here really formed in situ. I guess it is much more plausible – I particular given the current oligotrophic conditions and low TOC contents discussed above – that methane has migrated up from deeper sources.

Ls. 114 ff.: Also the argumentation in this sentence is odd. Even if waters are anoxic they almost always have the typical marine sulfate concentration of 28-30 mmol/l. Thus, anoxia does not necessarily lead to sulfate reduction.

Ls. 120 ff: The cores were sampled during cruises of R.V. Shikmona ...

Ls. 122 ff.; This sentence sounds odd. Please, rephrase.

L. 132: . . . the "stable carbon" isotopic composition . . . explain the abbreviation DIC

L. 134: "at" -20°C

L. 136: The wording in this sentence is a bit odd. Do you mean that the surface sediment has been lost during sampling (which is usually the case during gravity or piston coring)?

L. 137: Does it mean that you have sub-sampled the box corer by means of push cores? If yes, please say so.

L. 139: Does it mean that you have determined methane both in pore-water as well as sediment samples? How precisely and haw have the pore water and solid phase been separated?

Ls. 141 and 289: Some details of how precisely these incubation experiments have been performed are missing. How were the respective experiments/bottles killed? Did you use molybdate to inhibit sulfate reduction?

L. 143: Refer to the respective figure with pore-water profiles here.

L. 145: anoxic instead of anaerobic

L. 147: anoxically instead of anaerobically

L. 151: You are talking about mineral contents here – so the unit (mmol L-1) is not correct.

L. 152: In line 146 you have stated that incubations lasted for 3 months. Here you speak about 14 days?!

L. 161: It has to be "total sulfur" instead of sulfate. Sulfate can't be measured by ICP-AES

L. 162: has to be "inductively" and Perkin "Elmer"

At this point I stopped to correct typos and odd wording – there are just too many.

Ls. 16 ff.: a pore-water profile can't be "performed"; please also state which parameters have been analysed and in which figures they are shown; what do you mean with "and not their average"? This is absolutely unclear to me.

Ls. 170 ff.: I would suggest to insert a table, which gives the details of the extraction used – including reagents, solid-phase/reagent ratios, shaking times, etc.; please, also state whether the extractions has been performed on dry or wet sediment samples; if you used wet samples, how has porosity been determined? By the way, carbonate-associated Fe is not an "iron oxide" as stated at the beginning of this sentence.

Ls. 202 ff.: Again: pore-water profiles can't be performed. Please, rephrase.

Ls. 204 ff.: As also stated above you have not determined sulfate but total sulfur. So, rephrase accordingly and also correct this in Fig. 1 and throughout the manuscript.

L. 207: increase "with depth"

Ls. 207 ff.: I do not fully understand this sentence. Moreover, part of this sentence is interpretation/discussion and should thus not be part of the Results chapter.

Ls. 215 ff.: Large parts of this is discussion/interpretation.

Ls. 229: I found this sentence confusing because from the chapter 2.2 "Sampling" it

was not clear to me that the sites have been sampled three times. Please clarify and give a table summarizing the dates, exact positions, precise names etc. of the samples used in this study. What is the "Aug-13 core"? Where is it shown in Fig. 1? A legend and/or respective explanations in the figure caption are missing

L. 238: Why are deviating points not discussed?

Ls. 248 ff.: I can't find Fig. S1; solid-phase values are "contents" (not concentrations)

Ls. 257 ff.: A lot of this is already interpretation/discussion. Moreover, papers should not be cited in the Results chapter.

L. 303: Which station precisely do you refer to here? "at this station"? How do you know that intensive methanogenesis occurs in the respective sediment layer? Due to the fact that TOC contents in the shallow sediments are low and free gas is detected in deeper layers, I would rather suggest that methane is migrating up from the deeper subsurface. Please discuss and consider this carefully.

Ls. 305 ff.; This sentence needs to be rephrased.

Ls. 314 ff. and 331 ff.: As already stated above I do not agree that methanogenesis necessarily occurs in the respective sediment zone. To me it seems more likely that methane has migrated up from deeper layers.

Ls. 317, 351 and throughout the manuscript: What do you mean with iron oxide "reactivation"? This is odd.

Ls. 334 ff.: I do not understand at all how the findings link or relate to the Last Glacial Maximum?! How can the current environmental conditions be attributed to the Last Glacial Maximum or Mid-Pleistocene? You need to much more carefully discuss this.

L. 339: anoxic instead of anaerobic

Ls. 346 ff.: This has not been described in the respective methods chapter.

Ls. 351 ff.: And how does all of this relate to your data?

Ls. 358 ff.: Numerous papers that have discussed and presented evidence for Fe-mediated AOM in natural aquatic sediments have not been cited here.

Ls. 363 ff.: I would not overinterpret methane concentrations, which have been determined ex situ because methane typically suffers from strong degassing during core retrieval.

Ls. 412-415: These two sentences more or less say the same.

From the discussion, as it is presented, it is not clear to me at all which novel findings your study and data contribute to the discussion on and research topic of potential drivers of deep iron reduction.

---

## Author Comment (AC2) · 8 Apr 2019

"Evidence for microbial iron reduction in the methanogenic sediments of the oligotrophic SE Mediterranean continental shelf" by Vigderovich et al.

Response to comments from anonymous reviewer #2 (our response in blue):

This manuscript presents pore-water data (S, $CH_4$, $Fe^{2+}$, $H_2$, $d^{13}C$-DIC), results of incubation experiments as well as data on the abundance and diversity of bacteria and archaea in sediments of the South Eastern Mediterranean continental shelf. Besides a typical zone of organoclastic iron reduction observed close to the sediment surface the authors report a second zone of enhanced $Fe^{2+}$ pore-water concentrations within the methanic sediments below the sulfate/methane transition. Evidence for iron reduction in methanic subsurface sediments is commonly found in high accumulation continental shelf and margin sediments and a strong research interest currently exists in elucidating which (bio)geochemical pathways and potential microbial organisms mediate this "deep" iron reduction.

In this respect, the paper focusses on an important and topical research question and is in principle suitable for Biogeosciences. However, I regret to say that the manuscript has numerous flaws and appears as if it has not been prepared with the required care. The manuscript thus needs a major overhaul before I can recommend publication. The English also requires quite some polishing and I would suggest to ask an English native speaker to proofread the manuscript. There are numerous typos (which I have not all corrected in detail) and the wording is imprecise in many places – all this need careful checking and correction.

We thank the reviewer for the thorough and constructive review. We addressed and accepted all comments (see below) and revised the manuscript accordingly. In addition, we have edited and proofread the English.

Several issues that need to be considered when preparing a revised version:

1) The most important point is that the discussion is not adequate as it stands, several assumptions are not supported by the data and many key publications have not been cited. Often statements occur in the form of single sentences without "really" discussing the data obtained in the framework this study.

We accept this comment. In the revised version we have extended and strengthened the discussion (see below in the specific comments the clarifications and added calculations). We carefully analyzed the data, clarifying what is indicated directly and is supported by other publications (also listed below), and what is speculative.

2) It is not clear to me which novel findings your study contributes to the topic of deep iron reduction. This needs to be outlined precisely.

In the revised version we present and clarify the novel aspects of this paper:

 a. **Combining** geochemical profiles, microbial profiles and incubation experiments to show evidence for microbial iron reduction in the deep methanic zone and the potential microbial population performing this reduction.
 b. Showing that this deep iron reduction can occur even in sediments of oligotrophic seas, such as the **oligotrophic** SE Mediterranean. We suggest that the availability of iron minerals for reduction is linked to an intensive methane cycle (see below, addressing the comment on L. 111).

In the revised version we emphasize the connection between the deep iron reduction and the methane cycle more clearly.

3) Please provide a map that shows the study area and the three sampling locations and a table that summarizes the dates, exact positions, precise names etc. of the samples used in this study.

A map and a table were added to the revised manuscript.

4) Please, also add a table that gives the details of the sequential extractions performed in this study.

A table with the sequential extractions details was added to the revised manuscript.

5) Referencing is not adequate – i.e. several relevant papers are missing. I have listed some publications below but a careful literature search should be performed.

We thank the reviewer for this list. We added those references and several more. The following references were added (see below the specific places):

Boetius et al., 2000; Egger et al., 2017; Emerson et al., 1980; Hinrichs et al., 1999; Hoehler et al., 1994; Iversen and Jorgensen, 1985; Knittel and Boetius, 2009; Li et al., 2012; Lovley 1991; März et al., 2018; Milkov and Sassen, 2002; Milkov, 2004; Moutin and Raimbault, 2002; Niewöhner et al., 1998; Oni et al., 2015; Orphan et al., 2001; Paull et al., 2008; Riedinger et al., 2017; Wurgaft et al., 2019; Zhang and Lanoil, 2004.

6) Please, precisely distinguish between and separate Results and Discussion. The Results chapter already contains a lot of interpretation/discussion and several references, which is formally incorrect.

We accept this comment, and the results and discussion sections were properly separated, the data was presented first and then discussed. We also moved the references to the discussion section.

Specific comments:

Line 2 and throughout the manuscript: I do not like the term "methanogenic" very much because it implies that methane formation occurs in the respective sediment layer/interval. Based on your considerations on page 3 (lines 102 ff. and lines 112 ff.) concerning the current oligotrophic conditions in the study area as well as the deeper gas front detected based on seismic profiling, I suggest that it is likely that methane is diffusing/migrating up from deeper layers into the sediment depths investigated in this study. I would thus propose to speak of "methanic" sediments, which is more neutral.

We completely agree with the reviewer that some of the methane in the pore-water originates from deeper sediments. This is indeed an important factor in this system that was discussed in our previous studies (which did not focus on iron - Sela-Adler et al., 2015; Wurgaft et al., 2019). We clarified this point in the revised version and included flux calculations (see below, addressing the comment on L. 111). As suggested, we rephrased the term to "methanic".

L. 20: What exactly do you mean by "mechanistic" nature?

The microbial link between the iron and the methane cycles in marine sediments, either by competition between methanogens and iron reducing bacteria due to environmental

conditions, methanogens switching from methanogenesis to iron reduction metabolism or iron driven AOM. We explained this in the revised version.

L. 25: delete "cores"; in the deeper methanic zone

Deleted.

L. 27: Do you mean $Fe^{2+}$ concentrations in pore water?

Yes. Specified in the revised manuscript.

L. 37: Li et al. (2012) is only one of a vast amount of literature on this topic – you may add a few other papers. So change to (e.g. Li et al., 2012; Riedinger et al., 2017 (Frontiers in Earth Science); März et al., 2018 (Mar. Geol.).

We added all the relevant references. In addition to the ones above, we cited Egger et al., 2016; Ettwig et al., 2016; Sivan et al., 2014; Slomp et al., 2013. These references support the fact that Fe(III) minerals have a key role in the biogeochemical cycles of carbon, sulfur, phosphorous and nitrogen.

L. 45: What exactly do you mean with "outward" diffusing methane? This is not clear to me. Please, specify.

The term infers that the methane diffuses away from the methanic zone to the SMTZ or deep layers. In the revised manuscript this was clarified.

L. 47: Key papers on sulfate-mediated AOM are missing here: please add at least Hinrichs et al. (1999) and Boetius et al. (2000). . . . it should then read: (e.g. Hoehler et al., 1994; Hinrichs et al., 1999; Boetius et al., 2000) . . .. and you may of course add further papers.

As we do not focus on sulfate-mediated AOM, we did not include most works on this topic. However, we agree with the reviewer that at least the key works should be included. We added thus these references, as well as Orphan et al., 2001; Knittle and Boetius 2009.

L. 49: Also here Valentine (2002) is only one example of a vast amount of literature on this topic. You may also wish to cite Niewöhner et al. (1998), GCA, here.

Niewöhner et al. (1998) work from the west African margin was added.

L. 51: Has to be iron "reduction" (instead of oxidation)

Corrected.

Ls. 58/59: Please, give the respective references.

The reference was added (Lovley 1991)

L. 60: Please rephrase to: . . . incubation of marine seep sediment . . ..

The sentence was rephrased as suggested.

Ls. 61 ff.: Please also cite the following papers in this context: März et al. (2008), Oni et al. (2015), Egger et al. (2018), who have also presented evidence for Fe-coupled AOM in marine, coastal, and brackish sediments.

We accept this comment and have added these references (Marz et al. 2008; Oni et al., 2015; Egger et al., 2017) here as evidence for deep iron reduction.

Ls. 68 ff.: Please, also cite Oni et al. (2015) here who have presented microbial studies for the methanic zone of North Sea sediments.

Added, but in line 62 (in the original version), since the original line 68 is about freshwater sediments and Oni et al. (2015) studied the North Sea sediments. The original line 62 was rephrased to: "It was suggested through the modeling of geochemical profiles in deep sea sediments (Sivan et al., 2007; Marz et al., 2008; Riedinger et al., 2014), in microbial studies of marine sediments (Oni et al., 2015)…"

Ls. 74 ff.: This sentence is hard to follow and sounds a bit odd. Please, rephrase.

The sentence was rephrased to: "Whereas Fe(II) is highly soluble, Fe(III) that is the most abundant specie of iron under natural conditions, appears as low solubility minerals."

L. 79: I would not speak of "inactive" in this context but rather of "of low reactivity". Furthermore, I do not find it surprising that reactive iron oxides are preserved and present below the SMT. This finding has already been explained by several studies/papers – amongst others by Riedinger et al. (2005), GCA, März et al. (2008), Mar. Geol., and März et al. (2018), Mar. Geol.

The term "inactive" was changed to "of low reactivity" as suggested. We accept the comment and removed this word.

L. 87 ff.: You may also wish to cite Oni et al. (2015) here.

This sentence focuses on methanogenesis inhibition by iron reduction, and thus this reference was not added here. It can be found in other places in the revised manuscript (see above).

L. 92: What exactly do you mean with "reactivate" in this context? This is not clear to me – please specify. Were the Fe oxides "unreactive" before? By which process/condition have they been "reactivated"?

We infer that the iron oxides, which were not reduced in the upper sedimentary column by bacteria or archaea, are reduced in the deeper sediments, even though there is less energy for redox reactions. This suggests that there is some advantage at these depths that allows their reduction. Several processes may explain this reactivation: 1) Iron reducing bacteria succeed in outcompeting methanogens due to environmental changes, 2) the methanogens themselves switch to iron reduction due to some advantages (electron shuttling such methanophenazines?), or/and 3) the methane that is produced is more available for reduction than other organic substrates (Fe-mediated AOM). We clarified this point better in the revised version.

L. 97: What precisely is a "basic" incubation experiment?

We infer a fundamental incubation experiment. We removed the word basic and rephrased it in the revised version to: " We show both geochemical pore-water profiles and microbial investigation at three different stations combined with a simple incubation experiment with slurry…"

L. 99: Please, rephrase to: . . . possible links between the cycling of iron and methane".

Changed as suggested.

L. 102: I find it hard to imagine that the Levantine Basin is really one of the most oligotrophic marine settings in the world. I thought that globally the most oligotrophic ocean area is the South Pacific Gyre?! Please, check carefully and rephrase accordingly.

To the best of our knowledge, the Levantine basin is considered an ultra-oligotrophic marine system. For example, Thingstad et al. (Science, 2005) discussed the phosphorus imitation in the "Ultraoligotrophic Eastern Mediterranean", as well as several other studies, which ranked the Mediterranean basin as oligotrophic to ultraoligotrophic based on nutrients, chlorophyll a and PP pools (Krom et al., 1991; Antoine et al., 1995; Siokou-Frangou et al., 2010; Kress et al., 2014; references in Herut et al., 2016 and more). However, we rephrased the sentence to: "The Levantine Basin of the SE Mediterranean Sea is an oligotrophic nutrient-poor marine system (Kress and Herut, 2001)."

L. 109: I do not believe that the TOC contents are/were really "zero". I think this is an issue of the detection limit of the specific analytical method used. Please check.

Indeed, a typo mistake. We corrected the sentence: "… the Levantine Basin have low TOC levels of ~1% (~0.5 – 1.4%; Sela-Adler et al., 2015; Astrahan et al., 2017)."

Ls. 111 ff.: I do not understand the argumentation in this sentence. How can you conclude that methane found in shallow sediments is of biogenic origin if a deep gas front has been detected by seismics? Are you sure that the methane found in the shallow sediments investigated here really formed in situ. I guess it is much more plausible – I particular given the current oligotrophic conditions and low TOC contents discussed above – that methane has migrated up from deeper sources.

As written above, we agree that some of the methane has migrated from deeper sources, at least in Station SG-1. However, our results indicate that part of the methane is also produced *in-situ* in the methanic zone (zone 3) based on our geochemical profiles mainly of $\delta^{13}C_{CH4}$ and $\delta^{13}C_{DIC}$ (Sela-Adler et al., 2015), and the mcrA profile (presented here). The geochemical profiles show the transition from sulfate reduction to methanogenesis, a clear SMTZ, very low carbon isotopic value of the methane (between -80 and -100‰) and classical "*in-situ*" diffusive $\delta^{13}C_{DIC}$ profiles with the significant increase in the isotopic values below the SMTZ in the methanogenic zone. The microbial profile shows that the mcrA gene copy number increases with depth and peaks below the SMTZ. All fits to *in-situ* biogenic methane production in zone 3, in addition to some migration. We clarified this in the text, writing clearly the two sources of methane and their supporting evidence.

As mentioned above, we discussed the migration of methane in our previous studies. In Wurgaft et al. (2019) that focused on sulfate reduction rates in the SMTZ based on alkalinity and DIC profiles, we wrote: "The similarity between sulfate reduction rates in the ultra-oligotrophic Southeastern Mediterranean and these eutrophic regions suggests that "external" methane, which is not the product of degradation of organic material originating in the water column but rather derives from deeper deposits, provides an important source of reducing power to the SMTZ. Such deep methane deposits and upward fluxes are common in many continental margins (e.g. (Milkov and Sassen, 2002; Milkov, 2004; Paull et al., 2008; Zhang and Lanoil, 2004))".

We agree with the reviewer that this source may explain our results of the low TOC. In the revised manuscript we suggested and clarified that this source of methane leads to intensive sulfate-mediated AOM in the SMTZ, and that this intensive process and

biomass may serve as additional substrate that "fuels" the deeper zone, activating the iron-oxides. We added to the text the following part with the calculation of the biomass that is produced from this source: "The importance of methane flux as a carbon source that supports the deep microbial community in the sediments of the SE Mediterranean can be inferred by comparing the organic carbon flux from the photic zone, with the flux of organic carbon that is oxidized by sulfate in the pore water. Using sediment traps, Moutin and Raimbault (2002) estimated an export flux of $7.4\pm6.3$ mg C m$^{-2}$ d$^{-1}$, which leaves the photic zone there. However, Wurgaft et al. (2019) estimated that the flux of DIC entering the SMTZ from sulfate reduction is equivalent to $8\pm3$ mg C m$^{-2}$ d$^{-1}$. While the difference between the two fluxes is statistically insignificant, it should be noted that the flux of organic material that survives aerobic oxidation in the water column and the upper part of the sediment column, as well as anaerobic oxidation by other electron acceptors with higher energy yield (Emerson et al., 1980; Froelich et al., 1979), is likely to be substantially smaller than the flux measured by Moutin and Raimbault (2002). Therefore, it is unlikely that export flux from the photic zone constitutes the sole source of carbon to the SMTZ. Wurgaft et al. (2019) suggested that methane originating from deep sediments and migrating upwards in the pore-fluids provides an important source of carbon to the SMTZ in SG-1. Methane sources of such are common along continental margins sediments (e.g. Milkov and Sasson, 2002; Milkov, 2994; Paull et al., 2008; Zhang and Lanoil, 2004). Here, we suggest that the supply of methane leads to intensive sulfate-mediated AOM in the SMTZ, and that this process produces(??) biomass which may serve as additional substrate. (New sentence) that "fuels" the deeper zone, activating the iron-oxides."

Ls. 114 ff.: Also the argumentation in this sentence is odd. Even if waters are anoxic they almost always have the typical marine sulfate concentration of 28-30 mmol/l. Thus, anoxia does not necessarily lead to sulfate reduction.

We agree with the reviewer and clarified the sentence at the beginning of the study site section: " The bottom seawater across the continental shelf is well oxygenated and sulfate concentration in the water-sediment interface is ~30 mmol L$^{-1}$ (Sela-Adler et al., 2015)."

Ls. 120 ff: The cores were sampled during cruises of R.V. Shikmona ...

Corrected as suggested.

Ls. 122 ff.; This sentence sounds odd. Please, rephrase.

We rephrased this sentence to: "These stations were previously investigated for other purposes…"

L. 132: . . . the "stable carbon" isotopic composition . . . explain the abbreviation DIC

DIC- dissolved inorganic carbon. This abbreviation is explained in L 39.

L. 134: "at" -20∘C

Corrected.

L. 136: The wording in this sentence is a bit odd. Do you mean that the surface sediment has been lost during sampling (which is usually the case during gravity or piston coring)?

We refer to the uppermost sediment of the piston core, which is indeed usually mixed with the top seawater entrapped between the surface sediment and the piston. To

avoid any disorder in the surface sediment, we have used a box corer sub-sampled by Perspex push cores for the top ~30 cm sediments. We revised therefore the sentence to: "The uppermost sediments were collected using a 0.0625 m² box corer (Ocean Instruments BX 700 Al). Two ~30 cm sediment cores were sub-sampled using Perspex tubes during the September 2015 and January 2017 cruises."

L. 137: Does it mean that you have sub-sampled the box corer by means of push cores? If yes, please say so.

Revised, see above.

L. 139: Does it mean that you have determined methane both in pore-water as well as sediment samples? How precisely and how have the pore water and solid phase been separated?

We have measured the methane from the total wet sediment, by transferring the sediment sample immediately to a crimped bottle with 5 mL of NaOH and flushed with nitrogen. Then measured the methane in the headspace. We explained it in the revised version.

Ls. 141 and 289: Some details of how precisely these incubation experiments have been performed are missing. How were the respective experiments/bottles killed? Did you use molybdate to inhibit sulfate reduction?

We agree with the reviewer that additional details regarding the incubation experimental design were needed. The revised manuscript includes the specific information regarding the "killed" bottles (sediment killing via autoclave). Molybdate was not used in the experiment.

L. 143: Refer to the respective figure with pore-water profiles here.

We agree with the comment and the figure (Fig 1) was referred in the revised MS.

L. 145: anoxic instead of anaerobic

Changed.

L. 147: anoxically instead of anaerobically

Changed.

L. 151: You are talking about mineral contents here – so the unit (mmol L-1) is not correct.

We changed the units of the mineral content to grams in the revised MS and the final Fe(III) concentrations in mmol L$^{-1}$ units in brackets.

L. 152: In line 146 you have stated that incubations lasted for 3 months. Here you speak about 14 days?!

The sediment was incubated only with synthetic sea water without sulfate (in a 1:1 sediment:water ratio) for three months prior to the experiment. The experiment then began with the division of the slurry to the 60 mL bottles, the addition of more synthetic water (final sediment:water ratio of 1:3) and some manipulations (addition of iron oxides and H₂ to some treatments). In the revised MS we clarified this point better.

L. 161: It has to be "total sulfur" instead of sulfate. Sulfate can't be measured by ICP-AES.

Correct, ICP-AES measures total sulfur. Since sulfide was not detected in the samples (by Cline method) and these are marine samples, we assume that "total sulfur" here is actually sulfate. However, we changed the title to total sulfur and clarify its meaning in the revised version.

L. 162: has to be "inductively" and Perkin "Elmer" At this point I stopped to correct typos and odd wording – there are just too many.

Corrected.

Ls. 166 ff.: a pore-water profile can't be "performed"; please also state which parameters have been analysed and in which figures they are shown; what do you mean with "and not their average"? This is absolutely unclear to me.

The word "performed" was changed to "produced" in this context throughout the text. We agree with the reviewer that the other term is unclear, and it was removed.

Ls. 170 ff.: I would suggest to insert a table, which gives the details of the extraction used – including reagents, solid-phase/reagent ratios, shaking times, etc.; please, also state whether the extractions has been performed on dry or wet sediment samples; if you used wet samples, how has porosity been determined? By the way, carbonate associated Fe is not an "iron oxide" as stated at the beginning of this sentence.

We thank the reviewer for the suggestion, in the revised version a detailed table was added with the specifics of the extractions. The extractions were conducted on dry sediment. In addition, the word "oxides" was changed to "minerals".

Ls. 202 ff.: Again: pore-water profiles can't be performed. Please, rephrase.

Rephrased to "produced".

Ls. 204 ff.: As also stated above you have not determined sulfate but total sulfur. So, rephrase accordingly and also correct this in Fig. 1 and throughout the manuscript.

The reviewer is correct, ICP-AES measures total sulfur. We clarify this in the revised version as mentioned above.

L. 207: increase "with depth"

Corrected.

Ls. 207 ff.: I do not fully understand this sentence. Moreover, part of this sentence is interpretation/discussion and should thus not be part of the Results chapter.

We agree with the reviewer that the sentence is not clear, we also agree with the other comment and moved it to the discussion chapter. The sentence was rephrased to: "The maximum methane concentration was approximately 10 mmol L$^{-1}$ at ~140 cm depth…"

Ls. 215 ff.: Large parts of this is discussion/interpretation.

We agree and moved it to the Discussion chapter.

Ls. 229: I found this sentence confusing because from the chapter 2.2 "Sampling" it was not clear to me that the sites have been sampled three times. Please clarify and give a table summarizing the dates, exact positions, precise names etc. of the samples used in this study. What is the "Aug-13 core"? Where is it shown in Fig. 1? A legend and/or respective explanations in the figure caption are missing.

We agree with the reviewer that the study sites sampling time was not clear in the previous version. The stations SG-1 and PC-3 were sampled three times each during different cruises and station PC-5 was sampled once. The text was clarified and a table with the specifics was added.

L. 238: Why are deviating points not discussed?

The few deviations are of only one data point each, and are probably due to an analytical error during the measurement/sampling process. We clarify it in the revised version.

Ls. 248 ff.: I can't find Fig. S1; solid-phase values are "contents" (not concentrations)

Figure S1 can be found in the supplementary material, perhaps there was an error and the reviewer did not receive the file?

The word "concentrations" was changed to "content".

Ls. 257 ff.: A lot of this is already interpretation/discussion. Moreover, papers should not be cited in the Results chapter.

We agree, and moved part of it to the discussion, as well as the references.

L. 303: Which station precisely do you refer to here? "at this station"? How do you know that intensive methanogenesis occurs in the respective sediment layer? Due to the fact that TOC contents in the shallow sediments are low and free gas is detected in deeper layers, I would rather suggest that methane is migrating up from the deeper subsurface. Please discuss and consider this carefully.

We are referring to station SG-1. This is clarified in the revised version. We agree that some of the methane migrated up from deeper subsurface (see above). We rephrased the sentence to: "At station SG-1 methane reaches higher concentrations, which leads to intensive methane oxidation by sulfate at the SMTZ…"

Ls. 305 ff.; This sentence needs to be rephrased.

We agree and rephrased it to: "…causing it to occur at shallower depth and produce lower $\delta^{13}C_{DIC}$ values than the other two stations, as observed in previous studies (e.g. Sivan et al., 2007)."

Ls. 314 ff. and 331 ff.: As already stated above I do not agree that methanogenesis necessarily occurs in the respective sediment zone. To me it seems more likely that methane has migrated up from deeper layers.

As mentioned above, we now refer to the two sources.

Ls. 317, 351 and throughout the manuscript: What do you mean with iron oxide "reactivation"? This is odd.

Please see above.

Ls. 334 ff.: I do not understand at all how the findings link or relate to the Last Glacial Maximum?! How can the current environmental conditions be attributed to the Last Glacial Maximum or Mid-Pleistocene? You need to much more carefully discuss this.

We removed this sentence. The hypothetical environmental conditions are discussed by Sela-Adler et al., 2015 and Schattner et al., 2012, while not directly linked to this study.

L. 339: anoxic instead of anaerobic

Corrected.

Ls. 346 ff.: This has not been described in the respective methods chapter.

The reviewer is correct, the matter is elaborated in the revised version in the methods chapter: "One mL of $H_2$ was added by gas tight syringe to two bottles with addition of hematite and two bottles with addition of magnetite (to final concentration of ~4% of the Head space volume)."

Ls. 351 ff.: And how does all of this relate to your data?

Cryptic sulfur cycle is observed more and more in marine sediments (e.g. Holmkvist et al., 2011; Brunner et al., 2016). It seems that this cycle is possible here based on the microbial populations that contain those that may be involved in sulfur cycling (from 16S analysis). Also, pyrite was found in the methanogenic zone (Wurgaft et al., 2019). We clarify this point in the revised version.

Ls. 358 ff.: Numerous papers that have discussed and presented evidence for Fe mediated AOM in natural aquatic sediments have not been cited here.

As mentioned above, in the revised version we include the main literature on Fe-AOM.

Ls. 363 ff.: I would not overinterpret methane concentrations, which have been determined ex situ because methane typically suffers from strong degassing during core retrieval.

We agree with the reviewer and rephrased this sentence which now emphasizes just the general trend: " In our profiles AOM could be a valid option. As can be inferred from figure 5, some association was observed between the dissolved Fe(II) concentrations in zone 3 and the methane concentrations. It seems that at high concentrations of Fe(II), methane concentrations are low and vice versa."

Ls. 412-415: These two sentences more or less say the same.

We agree with the reviewer that the two sentences sound similar, however the first sentence is the key sentence of the paragraph, and the following three sentences are listing the main results of the study.

From the discussion, as it is presented, it is not clear to me at all which novel findings your study and data contribute to the discussion on and research topic of potential drivers of deep iron reduction.

Please see above.

---

## Author Response (AR2)

"Evidence for microbial iron reduction in the methanogenic sediments of the oligotrophic SE Mediterranean continental shelf" by Vigderovich et al.

We would like to thank the editor and the two reviewers for their supportive and constructive comments again.

Response to comments from anonymous reviewer #1 (our response in blue):

We thank the reviewer for the positive review and address below the few technical issues that were left.

L28 and Figure 3: italicize mcrA
Was corrected.

L89, L373: add hyphen between "iron" and "reducing"
Was added.

L100-L101: there is a grammatical error here that needs to be fixed: "marine's methanic zone"
Was corrected.

L105: add a subject and verb to sentence: "This _____ is ____ by...:"
The sentence was revised to: "The microbial iron reduction is observed by using both geochemical…"

L313: the italics of "uncultured Bathyarchaeota" are different than all the other phyla listed -- format consistently
Was corrected.

Response to comments from anonymous reviewer #2 (our response in blue):

The manuscript by Vigderovich represents the revised version of a manuscript that I have reviewed previously. Several of the issues pointed out in my previous review have been addressed by the authors. However, there are still some points and issues that need attention. It is striking that there are still many imprecise statements and numerous typos. I have corrected only a few (see specific comments below). In particular, the English still needs some substantial overhaul and careful polishing and check by a native speaker. Also some of the figures/plots (in particular Fig. 2 and caption) are insufficiently labeled and references cited in the text are missing in the list of references.

We thank the reviewer for the detailed and constructive review again. We are sorry that there were still typos, as the manuscript had been edited twice. We corrected the additional comments, and the ms was edited again by native English colleague. We will be glad to send the revised version to additional editing of the journal if needed.

Besides these formal flaws the main points are: 1) The Discussion chapter still has many statements that are much too general and it is not clear how these relate to your data and to previous work. References are often missing to support the statements made;

We accept this comment and improved the discussion chapter and corrected the statements according to the specific comments below.

2) Please expand and specify and discuss in detail your geochemical evidence showing the contributions of methane from deeper sources (the „gas front") and that being produced in situ in the shallow sediments.

We accept this comment and expended our discussion on the source of the methane.

To conclude, I find this study and manuscript very interesting and definitely suitable for Biogesciences. However, it needs another intense round of clarification of the scientific discussion and formal polishing before I can recommend publication.

We thank the reviewer again, and improved the manuscript according to the specific comments below.

Specific comments

L. 28: microbially mediated

Corrected

Ls. 33/34: I do not fully understand this last sentence of the abstract. What do you mean with „deeper microbial activity" and „methanic iron reduction"? I also do not agree with the statement that (what I think you suggest) Fe reduction driven by methane oxidation is observed. Of course you observe Fe reduction within the methanic zone, which leads to liberation of Fe2+ into the pore water. Please, rephrase and specify accordingly.

By saying "deep microbial activity" we mean the microbial activity deeper than the sediment-water interface. In addition, we did not intend to state that there is iron-coupled AOM here (this possibility is discussed in the discussion chapter). We clarified the sentence to: "We suggest that intensive upward migration of methane in the sedimentary column and its oxidation by sulfate may fuel the microbial activity in the SMTZ. The biomass, created by this microbial activity, can be used further below by the iron reducers at the methanic zone in the sediments of the SE Mediterranean."

Ls. 38, 40: add e.g. in brackets and also cite work by Lovley et al. here who have performed key studies on microbial Fe reduction.

The brackets and the following references were added: Lovley and phillips, 1986; Lovley et al., 1987; Lovley and phillips, 1988; Lovley 1997.

L. 57: add e.g. in brackets; Niewöhner et al. (1998) is missing in the list of references.

Sorry, it was somehow by mistake deleted, and now added.

L. 54: add Hoehler et al. (1994) here

Sorry, it was again somehow by mistake deleted, and now added.

Ls. 68 ff: Sentence is odd. Please rephrase. Furthermore, not all of the studies cited here, have performed modelling. Please, check carefully and correct accordingly.

The sentence was rephrased to: "This process in marine sediments was shown through incubation experiments in marine seeps sediments (Beal et al., 2009; Sivan et al., 2014). It was also suggested to exist mainly through geochemical profiles in deep sea sediments and their modeling (Sivan et al., 2007; März et al., 2008; Riedinger et al., 2014)…"

L. 81: … „under" natural conditions ….

Corrected

L. 100: I would suggest to rephrase to: „Despite …. the link between the biogeochemical cycling of iron and methane in the methanic zone of marine sediments …."

Was rephrased as suggested.

L. 103: … we report the observation of microbial iron reduction ….

Corrected.

L.105: What are „microbial profiles"? Please specify – profiles of what precisely?

The microbial profiles are the 16S rRNA and qPCR 16S of bacteria and archaea and the functional gene mcrA. This is specified in the revised version.

L. 116: is composed of

Corrected.

L. 121: „contents" inspite of levels

Was changed.

L. 123: I do not agree that TOC contents of more than 1 wt% - as given here – can be described as low or underlying oligotrophic areas. I find this rather high. Please, also specifiy whether you speak of trophic level of the surface waters (e.g. oligotrophic) or TOC contents of the sediments.

We revised the sentence to clarify that the TOC levels in sediments of the Levantine are lower than in the western basin and the Nile delta. Thus, the term "relatively low" was removed. We also clarified and referred on the oligotrophic nature of the surface water (photic zone) in line 114 (line 119 in the revised MS). It should be noted that the surface water certainly accounts as oligotrophic.

Ls. 125 ff. and 330 ff.: The statement and description of the gas front appear rather imprecise to me. What kind of gas front are you referring to? Why do you put it in brackets? Do you mean free gas? At which sediment depth was this gas front/free gas found. Furthermore, it is not clear to me whether the methane samples you describe as being of biogenic in origin come from this gas front depth or below the gas front or from shallower sediments overlying this gas front. If yes, what is then the link? Does the shallow methane analysed originate from the deeper sediments below the described gas front? Please specify.

We are referring to the gas front that was found and described by Schattner et al. (2012). The gas front is the top of a free gas zone in the sediment, which is limited to the shelf in area of about 72 $km^2$ from few to tens meters below the sea floor. The origin of the gas was speculated in Schattner et al. (2012), however it has not been sampled for isotopic values. The methane that was sampled by us was from the sediments overlying this gas front (~1-5 m depth). This shallow sediment methane is probably biogenic based on its low $\delta^{13}C_{CH4}$ values and the high C1/C2 ratio (Sela-Adler et al., 2015). The microbial and geochemical pore-water profiles indicate also that at least part of it is produced *insitu*. However, we do not know how much of the methane in the shallow sediments is originated from the gas front and how much of it is produced *insitu*. This was clarified and added to the text.

L. 130: delete „seafloor"

Was deleted.

L. 135: What do you mean with „methanogenesis characteristics"?

The sentence was rephrased to: "… and the possibility for methanogenesis to occur…"

Ls. 136 ff. and 171 ff.: How was porosity determined? This is needed to calculate methane concentrations in pore-water.

Porosity was determined by drying at 60℃ wet sediment samples from different depths, until there was no weight loss (~48 hr). The porosity was calculated as the weight loss from the initial weight of the samples. The porosity was indeed considered in the methane concentration in the pore-water calculations. This was added to the text.

L. 137: „anoxic" instead of anaerobic

Corrected.

L. 153: apparant „from (instead of in)

Changed as suggested.

L- 156: … to reach „a" 1:1 …..

Corrected.

L. 176: with „a" detection limit

Corrected.

Ls. 179/180: … the measured total sulfur concentrations in „pore water" were ….

The phrase "pore-water" was added.

L. 182: delete „Several"

Was deleted

Ls. 189 ff.: How was the sediment dried? Was it freeze-dried. Please specify.

Dried in the oven at 60℃. We added this to the text.

Ls. 195 ff.: Where/in which figures are these data - i.e. the results of sequential extraction – plotted. At least I could not find them in Fig. 2.

It is the first graph on the right-hand side. We revised the figure to clarify it (graphs (f) and (l) in the revised version).

L. 196: The profile of pyrite was taken from Wurgaft et al. (2019).

The citation was there, but the sentence was changed as suggested.

Ls. 222 ff. This sentence needs tob e rephrased because the syntax is odd. How can you investgate the „source" of a process? This makes no sense.

The sentence was rephrased to: "… in order to characterize the iron reduction process in the methanic zone of the SE Mediterranean continental shelf and to identify its potential sources."

Ls. 224: The next sentence also makes no sense. How can several profiles show complete depletion of total sulfur at one station?! (You say: The pore-water profiles ….)

The sentence was rephrased to: "The pore-water profiles at Station SG-1 (Fig. 2) show complete depletion of total sulfur at approximately 150 cm depth in all extracted cores."

L. 231 and throughout the manuscript: I do not like the term „traditional" iron reduction zone. This is imprecise and makes no sense. Better speak of the „upper Fe reduction zone" or „upper iron-rich zone".

The term was changed as suggested.

L. 234: „sediments" instead of sediment cores.

Changed.

L. 238 ff.: Are these methane concentrations in pore water? i.e. have the measured values been corrected for porosity? Please specify!

Yes, these are methane concentrations in the pore-water after correction to the porosity (this was added to the methods section in the revised MS).

L. 249: resemble

Corrected.

L. 256: I would not speak of „iron mineral profiles" but of „operationally defined iron mineral fractions"

Changed as suggested.

L. 266: I thought the pyrite profile was from Wurgaft et al. (2019)?! If yes, please cite this study as the source of these data here.

Was cited.

L. 267 and throughout the manuscript: use „uppermost" instead of „first" etc.

Changed as suggested where the uppermost part was the correct term.

Ls. 320 ff. and Fig. 2: The plots and profiles referred to here and depicted in Fig. 2 are insufficiently labeled. It is not clear to me at all what is shown in Fig. 2. Label the individual plots with a,b,c …. And refer to it in the figure caption and label the sequentially extracted Fe fraction as done in related publications.

Labeled as suggested.

L. 317 ff: This contradicts your statement in lines 223 ff. where you are mentioning „sources". Please precisely present the objective of your study.

The statement in line 223 was rephrased, please see above comment.

L. 321, last sentence, and Ls. 152 ff.: This belongs to the Materials and Methods chapter. Moreover, I would suggest to give a table in which the experimental set-up of the different slurry incubation experiments/vials is listed – otherwise it is very hard to follow.

The sentence was deleted. In addition, a table with the experimental set up was added to the text.

Ls. 334 ff. and 495ff.: I have no idea which of your data show that and to what extent the „shallow sediment processes" studied in this contribution are linked to the deeper gas reservoir. How do you know and show that part of the methane you have analysed in the shallow sediments originates from the gas front (whatever that is) and from methanogenesis in situ? Please expand and specify and discuss in detail your geochemical evidence showing the contribution of methane from deeper sources (the „gas front") and that being produced in situ in the shallow sediments. I find it hard to believe that methanogenesis is really possible in these low-TOC (oligotrophic) deposits. If the sediments were really so low in reactive TOC as suggested by the title, I find it hard to believe that methanogenesis can happen in situ at all. I find it more plausible that the methane you have detected in the shallower sediments originates from deeper sources.

Please see the response to the comment on L. 125 regarding the gas front. The TOC levels were ~0.8% in Station SG-1 and ~1% in PC-3. These levels can support *insitu* methanogenesis (e.g. Sivan et al., 2007) in addition to the methane immigration from the gas front. We clarified this point in the revised version.

Ls. 344 ff.: See previous comment!

Please see above.

L. 349: Do you mean pore-water Fe here? If yes, please say so.

Yes. Was specified in the revised version.

Ls. 348 – 351: These statements are much too general and it is not clear how this relates to your data.

We described the data, referred to the figures and removed the general statements regarding the hydrogen and the iron (see the response to L 425).

Ls. 356 ff: Also this part is much too general and not at all supported by references. As has been shown by numerous studies, pore-water Fe is often below detection limit at the SMT due to pyrite formation (e.g. Riedinger et al., 2005, GCA, and 2017)

We agree that this is often the case at the SMT, however, in these lines we are discussing our specific results from the shelf of the SE Mediterranean Sea, where in a few profiles we see low concentrations of dissolved Fe(II) in the pore-water, for example, in the profile from June 2015 at SG-1 station.

Ls. 358 ff.: The only study cited here is the one by Whiticar (1999). Please, give more recent ones as well. There is an enormous amount of studies and literature on this issues published in recent years.

The references Holler et al. (2009) and Conrad (2005) were added.

Ls. 370/371: Please also add Oni et al. (2015) here.

Was added.

Ls. 375/376: This part of the sentence is odd. How can organic matter be formed from upward migrating methane at the SMTZ?

We accept the reviewer's comment and rephrased to:"… where it is produced by the microorganisms that live there and benefits from the upward migrating methane."

L. 380: As already pointed out in a previous comment. I do not find TOC contents of more than 1 wt% particularly low.

We removed the sentence.

L- 381: Again, how do you know how much of the methane is produced in situ and how much is coming from deeper sources? Please discuss.

Please see discussion to comment in L. 125. We do not have a way to quantify the exact relative contribution of each source and we clarify it in the text.

Ls. 384 and 401: What kind of „biomass formation" at the SMTZ are you referring to here? Again there are no references at all.

We are referring to the microbial community that live at the SMTZ, including ANMEs and sulfate reducing bacteria. The reference of Boetius et al. (2000) was added.

L. 386: To which „deep microbial community" are you referring to here? Do you mean those at the depth of the „gas front" or that in your shallower sediments? This is not clear here.

The "deep microbial community" is referred to the community at zone 2 and 3 (i.e. SMTZ and methanic zone). We agree that the sentence is not clear, and it was revised to: "The importance of the methane flux as a carbon source that supports the microbial community at zone 2 and 3 in the sediments of the SE Mediterranean…"

Ls. 399 ff.: Please also add Fischer et al. (2013), Nature Geoscience here.

Was added

Ls. 409 and 412: Please also cite Riedinger et al. (2005) and (2014) here. They were the first to highlight these environmental/depositional prerequisites.

Were added.

Ls. 425 ff.: Discuss why the H2 levels are that particularly high at these study sites.

Hydrogen levels are dictated from its sink (i.e. sulfate reduction, methanogenesis, iron reduction) and sources processes (mostly fermentation). It is usually found in low steady state concentration in the deep sediments. High concentrations can be explained by fermentation being the dominant process (regarding the hydrogen levels). At station PC-3 the $H_2$ levels do not increase in the methanic zone, which means that there is $H_2$ consumption. At station SG-1 the concentrations are lower than PC-3, meaning that in general the $H_2$ consuming processes are more intensive than in PC-3. At station SG-1 there is a maximum peak in the methanic zone, meaning that the production of $H_2$ is higher at that depth compared to the SMT zone, however it doesn't necessarily mean that $H_2$ is not being consumed. This was added shortly to the text instead of the general statement.

Ls. 470: Which evidence makes you „believe" that – i.e. makes you make this assumption. Please discuss.

We assume that the microbial characterization in this station is representative, based on preliminary low resolution (not published) measurements in other stations using old classifications. However, we agree that the word "believe" here does not fit and we removed the sentence.

[revised manuscript text omitted]